# ATOM OF UNDERSTANDING: INFORMATION-THEORETIC DECOMPOSITION FOR INTERPRETABLE 3D QUESTION ANSWERING

## ABSTRACT

3D Question Answering remains largely opaque, with existing approaches functioning as black boxes that provide no insight into how different modalities contribute to spatial reasoning. This lack of interpretability limits trust and understanding in critical applications like robotics and autonomous systems. We present ATOM (**A**daptive **T**ask-aware m**O**dular **M**odel), the first information-theoretic framework that operationalizes Partial Information Decomposition (PID) to achieve fully interpretable 3D question answering. ATOM explicitly decomposes multimodal interactions into four theoretically-grounded information atoms: point cloud uniqueness, image uniqueness, redundancy (shared cross-modal information), and synergy (emergent complementary information). Our framework provides transparent reasoning through a Query-driven View Aggregator for geometrically consistent visual features, a Contextual Grounding Module for description-guided visual grounding, a Question-aware PID module with theoretically-grounded regularization, and a Dynamic Atom Modulation mechanism that provides direct, quantifiable interpretability of each atom's contribution. Extensive experiments on ScanQA and SQA3D datasets demonstrate that ATOM achieves the performance comparable to prior work and, to the best of our knowledge, are the first to enable transparency in 3D reasoning. Our analysis reveals that different question types systematically rely on distinct information patterns that align with human spatial cognition.

## 1 INTRODUCTION

3D Question Answering (3D QA) has emerged as a central benchmark for embodied multimodal intelligence across robotics (Driess et al., 2023; Huang et al., 2023), immersive technologies (Ding & Chen, 2025), and autonomous driving (Qian et al., 2024; Marcu et al., 2024). Unlike traditional 2D QA using homogeneous images, 3D QA integrates heterogeneous data, including point clouds, multi-view images, depth maps, and natural language for spatial reasoning. For instance, while a 2D system can identify a "television" in an image, a 3D QA system can answer "Can I see the television from my position on the bed?" by reasoning about complex spatial relationships and occlusions in three-dimensional space. However, current research encounters two interconnected challenges: achieving robust performance in complex 3D environments and providing interpretable reasoning about how different modalities contribute to answers. Without understanding the relationship between geometric cues and visual information, models may produce unreliable results by exploiting spurious patterns while overlooking crucial spatial evidence.

Several approaches address these performance gaps. ScanQA (Azuma et al., 2022) fuses 3D geometry with language for object-grounded spatial reasoning, while SQA3D (Ma et al., 2022) introduces contextual agent pose descriptions for first-person perception. DSPNet (Luo et al., 2025) proposes a dual-vision architecture integrating multi-view images with point clouds via text-guided fusion, achieving state-of-the-art results. However, these advances come with a critical limitation: existing 3D QA frameworks function as black boxes, offering little insight into how modalities influence predictions (Wu et al., 2024; Joshi et al., 2021). As Fig. 1 shows, when asked "Can I see the television from my position on the bed?", black-box models may hallucinate by over-relying on language priors while neglecting geometric cues (Wang et al., 2024). This reveals both the opacity of current

approaches and the lack of understanding whether various sources provide complementary, redundant, or conflicting information (Baltrušaitis et al., 2019; Liang et al., 2023a).

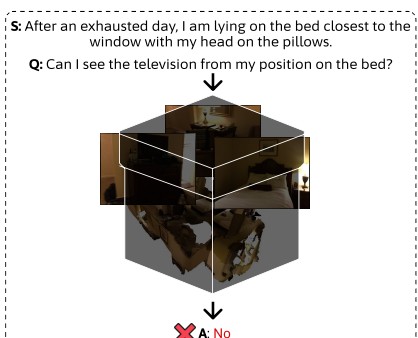 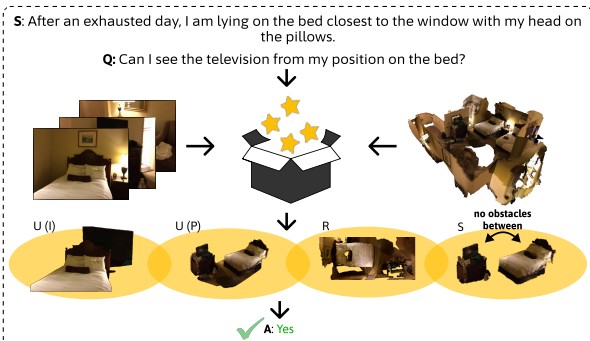

Figure 1: Traditional 3D QA models act as black boxes (**left**) and may hallucinate by over-relying on spurious cues. ATOM (**right**) addresses this by explicitly disentangling four information atoms: two uniqueness atoms (point cloud and image), redundancy and synergistic information, enabling a more interpretable multimodal reasoning.

Consequently, understanding multimodal interactions is key to advancing 3D QA. Partial Information Decomposition (PID) (Liang et al., 2023a; Wollstadt et al., 2023) provides a principled framework for this understanding by decomposing mutual information into four information atoms: two *uniqueness* atoms, each capturing modality-specific evidence, *redundancy*, the shared cross-modal information; and *synergy*, the complementary information emerging from their combination. By making these roles explicit, PID offers interpretability into how modalities influence predictions and establishes a systematic understanding of whether information sources overlap, diverge, or complement each other.

However, while PID offers theoretical insights into modality interactions, its integration into end-to-end 3D QA models remains unexplored. Existing fusion approaches do not explicitly model multimodal interactions (Azuma et al., 2022; Luo et al., 2025; Ma et al., 2022), instead, they focus on shared embedding spaces that align modalities for joint reasoning while overlooking distinct information atom roles. In contrast, recent multimodal frameworks attempt to implement PID concepts: CoMM (Dufumier et al., 2024) implicitly learns these interactions via contrastive objectives, but the resulting decomposition becomes entangled within monolithic representations, limiting interpretability. I²MoE (Xin et al., 2025) employs specialized mixture-of-experts with weakly supervised losses, yet suffers from training instability and lacks the question-aware decomposition critical for spatial reasoning tasks. Furthermore, both approaches target generic multimodal scenarios and fail to address the unique challenges of 3D QA: integrating geometric consistency across multi-view images, grounding language queries in 3D spatial contexts, and reasoning about object relationships in complex scenes.

We present **ATOM**, an information-theoretic framework for interpretable 3D QA. ATOM operationalizes PID in an end-to-end architecture where each component addresses specific multimodal reasoning stages. First, the Query-driven View Aggregator (QVA) draws visual features from geometrically consistent and question-relevant views. The Contextual Grounding Module (CGM) injects description context for stable spatial grounding. Building on these grounded representations, the Question-aware PID (Q-PID) explicitly decomposes them into four information atoms: point cloud uniqueness, image uniqueness, redundancy, and synergy (See Fig. 1), while atom-specific regularization losses enforce consistency with PID's theoretical definitions. Finally, the Dynamic Atom Modulation (DAM) adaptively reweights atoms based on questions, producing interpretable representations supporting robust reasoning.

The proposed framework, ATOM, is validated on the ScanQA (Azuma et al., 2022) and SQA3D (Ma et al., 2022) datasets. The results demonstrate that the proposed ATOM achieves competitive performance while providing interpretability through information-theoretic decomposition. Our main contributions are: **(1)** ATOM serves as the first effort of operationalizing PID for a 3D QA framework through question-aware information decomposition, maintaining state-of-the-art performance while

providing principled interpretability. **(2)** We introduce QVA for spatially consistent and question-relevant visual aggregation, CGM for description-guided grounding, and novel regularization losses that guarantee theoretical consistency with PID definitions in end-to-end training. **(3)** We develop DAM that enables question-aware atom weighting and comprehensive interpretability analysis at the sample and dataset levels, revealing how information atoms drive 3D spatial reasoning.

## 2 RELATED WORK

**3D Question Answering**  encompasses two primary settings: 3D visual question answering (3D VQA) (Azuma et al., 2022), which answers questions about 3D scenes from an external perspective, and 3D situated question answering (3D SQA) (Ma et al., 2022), which incorporates agent positioning and pose for first-person perspective reasoning. For 3D VQA, ScanQA (Azuma et al., 2022) designed fusion modules with transformer layers (Vaswani et al., 2017) to integrate word representations with 3D object proposal features for answer prediction. SQA3D (Ma et al., 2022) extended this by utilizing shared-parameter text encoders for situation descriptions, sequentially fusing object proposals with both situations and questions. Recent advances have leveraged transfer learning and architectural innovations: 3D-VisTA (Zhu et al., 2023) constructed a large-scale scene-text paired dataset, ScanScribe, and employed self-supervised pre-training with masked language modeling, masked object modeling, and scene-text matching to learn robust 3D-text alignments. Multi-CLIP (Delitzas et al., 2023) adapted CLIP's (Radford et al., 2021) contrastive learning framework to align 3D visual features with textual embeddings in shared representation spaces. DSPNet (Luo et al., 2025) introduced a dual-vision architecture integrating point clouds with multi-view images through Text-guided Multi-view Fusion (TGMF) and Adaptive Dual-vision Perception (ADVP), using Multimodal Context-guided Reasoning (MCGR) for cross-modal interaction and achieving current state-of-the-art performance.

However, these approaches face fundamental limitations in interpretability and reasoning design. Unlike 2D VQA, where attention mechanisms (Xu et al., 2015; Anderson et al., 2018) provide some insight into model decisions, 3D QA models remain largely opaque about how different modalities contribute to spatial reasoning. Most methods rely on black-box fusion processes that offer no principled understanding of multimodal interactions. Furthermore, both DSPNet (Luo et al., 2025) and 3D-VisTA (Zhu et al., 2023) concatenate description and question in SQA, conflating two fundamentally distinct reasoning processes. While scene grounding requires understanding spatial relationships, question reasoning demands logical inference, yet these methods treat them as single joint embeddings. These limitations necessitate principled frameworks that can explicitly decompose and interpret multimodal interactions in 3D spatial reasoning. ATOM addresses these challenges through structured information decomposition, providing transparent understandings of how geometric and semantic modalities interact for spatial reasoning.

**Multimodal Information Theory**  Modern multimodal learning primarily relies on contrastive methods (Radford et al., 2021; Jia et al., 2021) that align modalities into shared embedding spaces, yet these approaches are fundamentally limited in capturing redundant cross-modal information while overlooking modality-specific *uniqueness* and emergent *synergy* (Liang et al., 2023a). Information theory offers principled alternatives: mutual information estimation (Belghazi et al., 2018) and

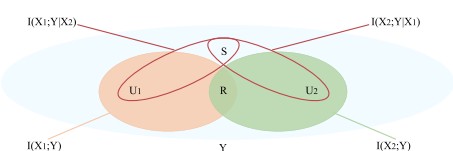

Figure 2: PID decomposes the total information $I(\{X_1, X_2\}; Y)$ into redundancy $\boldsymbol{R}$, uniqueness $\boldsymbol{U_1}$ and $\boldsymbol{U_2}$, and synergy $\boldsymbol{S}$.

information bottleneck principles (Alemi et al., 2016) have provided insights into representation learning, while Partial Information Decomposition (PID) (Bertschinger et al., 2014) specifically addresses multimodal interaction understanding. Given modalities $X_1$, $X_2$ and target $Y$, PID decomposes total mutual information by optimizing over distributions $\Delta_p$ that preserve marginals $p(x_i, y)$ while varying inter-modality coupling (Liang et al., 2023a) (See Fig. 2):

$$\boldsymbol{R} = \max_{q \in \Delta_p} I_q(X_1; X_2; Y) \quad \text{(redundancy)} \tag{1}$$

$$\boldsymbol{U_1} = \min_{q \in \Delta_p} I_q(X_1; Y|X_2), \quad \boldsymbol{U_2} = \min_{q \in \Delta_p} I_q(X_2; Y|X_1) \quad \text{(uniqueness)} \tag{2}$$

$$\boldsymbol{S} = I_p(X_1, X_2; Y) - \min_{q \in \Delta_p} I_q(X_1, X_2; Y) \quad \text{(synergy)} \tag{3}$$

This decomposition provides theoretical foundation for understanding whether modalities provide overlapping, unique, or complementary information, addressing black-box fusion limitations. While recent frameworks like CoMM (Dufumier et al., 2024), I²MoE (Xin et al., 2025), and FactorCL (Liang et al., 2023b) attempt to operationalize PID concepts, they suffer from critical limitations: entangled representations that obscure individual interactions, training instability from expert competition, and crucially, a lack of question-aware decomposition essential for spatial reasoning tasks. ATOM addresses these limitations through explicit question-aware PID decomposition and dynamic atom modulation, representing the first principled information-theoretic framework designed specifically for 3D question answering.

## 3 METHODOLOGY

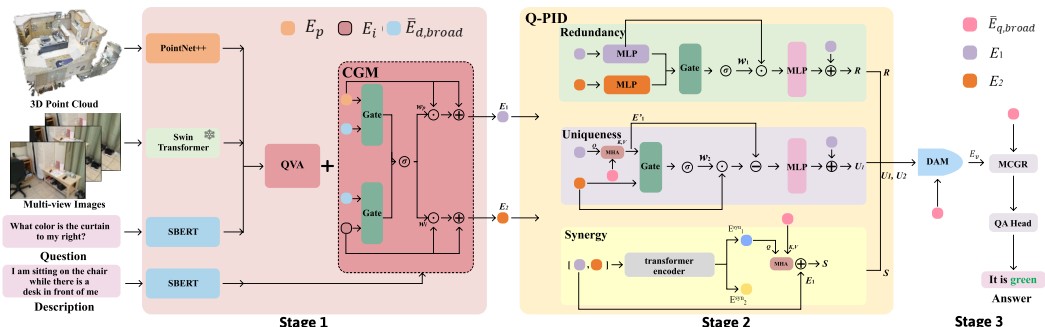

Figure 3: Overview of the proposed ATOM framework. ATOM operates in three stages: **Stage 1** extracts and grounds multimodal features using Query-driven View Aggregator (QVA) with optional Contextual Grounding Module (CGM) for situated QA tasks. **Stage 2** decomposes features into four information atoms (point cloud uniqueness, image uniqueness, redundancy, synergy) via Question-Aware PID (Q-PID). **Stage 3** applies Dynamic Atom Modulation (DAM) to adaptively weight atoms based on question guidance, creating interpretable representations that interact with question embeddings through MCGR (Luo et al., 2025) before final QA prediction.

We introduce ATOM (**A**daptive **T**ask-aware m**O**dular **M**odel), a novel framework that implements Partial Information Decomposition (PID) theory for interpretable 3D question answering. Given a 3D point cloud $\boldsymbol{P}$, multi-view images $\boldsymbol{I}$, and a natural language question $\boldsymbol{q}$ (optionally paired with a description $\boldsymbol{d}$ for SQA), ATOM predicts the answer $\hat{\alpha}$. Our framework operates through three key stages (Fig. 3): first, we extract and ground visual features using Query-driven View Aggregator (QVA) with optional Contextual Grounding Module (CGM) for situated scenarios; second, we decompose these features into four information atoms (point cloud uniqueness, image uniqueness, redundancy, synergy) via Question-aware PID (Q-PID); finally, we employ Dynamic Atom Modulation (DAM) to adaptively weight atoms based on questions, then interact the resulting interpretable representations with question embeddings through MCGR (Luo et al., 2025) for final answer prediction. To ensure faithful information decomposition, we introduce PID-aligned regularization losses and diversity constraints to prevent atom collapse.

### 3.1 STAGE 1: MULTIMODAL FEATURE EXTRACTION

We extract unified multimodal representations using specialized encoders (detailed in App. C.1) and map all features to a unified dimension $D_f$. Specifically, PointNet++ (Qi et al., 2017) processes the 3D point cloud $\boldsymbol{P}$ into $N_p$ seed features $\boldsymbol{E}_p$; a frozen Swin Transformer (Liu et al., 2021) encodes $M$ multi-view images into raw features $\boldsymbol{F}_i^{\text{raw}}$; and SBERT (Reimers & Gurevych, 2019) produces embeddings for both questions ($\boldsymbol{F}_q$) and scene descriptions ($\boldsymbol{F}_d$). We also derive global text representations $\bar{\boldsymbol{E}}_q$ and $\bar{\boldsymbol{E}}_d$ by pooling over tokens. Unlike prior works (Zhu et al., 2023; Luo et al., 2025) that concatenate the description and question into a single text input, thereby conflating situational grounding with reasoning, we preserve them as separate inputs to enable precise alignment and prevent cross-modal redundancy. As a result, our modules CGM (Sec. 3.1) applies description context

in visual grounding, while DAM (Sec. 3.3) adaptively weights information atoms based on question-specific reasoning requirements. Such targeted conditioning alleviates irrelevant information, thus grounding focuses purely on spatial cues while reasoning centered on logical inference.

**Query-driven View Aggregator (QVA).** To create point-aligned image representations, QVA aggregates multi-view features that are geometrically consistent with 3D points and further conditioned on the question for task relevance. We start by back-projecting each of the $N_p$ seed points onto all $M$ camera views, sample image features from $\boldsymbol{F}_i^{\text{raw}}$, and mask occluded or out-of-frame points, yielding valid multi-view features $\boldsymbol{F}_{\text{valid}} \in \mathbb{R}^{N_p \times M \times D_i}$. A linear layer $\phi_i$ maps them into the unified space, $\boldsymbol{I}_{\text{valid}} = \phi_i(\boldsymbol{F}_{\text{valid}}) \in \mathbb{R}^{N_p \times M \times D_f}$. The optimal view selection requires both spatial identity for geometric consistency and question context to filter irrelevant evidence and focus on task-relevant information. Accordingly, QVA generates point-specific queries by combining the point embeddings $\boldsymbol{E}_p$ with a broadcasted global question vector $\bar{\boldsymbol{E}}_{q,\text{broad}} \in \mathbb{R}^{N_p \times D_f}$. Specifically, we employ an MLP to process the concatenated point embeddings and broadcasted question vector, where $[\cdot, \cdot]$ denotes concatenation along channel dimension ($\boldsymbol{F}_{\text{query}} = \text{MLP}([\boldsymbol{E}_p, \bar{\boldsymbol{E}}_{q,\text{broad}}])$). We then use multi-head attention (MHA) to aggregate per-point, question-conditioned evidence across valid views: $\boldsymbol{E}_i = \text{MHA}(\mathbf{Q} = \boldsymbol{F}_{\text{query}}, \mathbf{K} = \boldsymbol{I}_{\text{valid}}, \mathbf{V} = \boldsymbol{I}_{\text{valid}}) \in \mathbb{R}^{N_p \times D_f}$. These represent as our image embeddings.

**Contextual Grounding Module (CGM) for SQA.** For SQA tasks, visual features ($\boldsymbol{E}_p$ and $\boldsymbol{E}_i$) require grounding in the description. To avoid the grounding–reasoning conflation in prior works (Luo et al., 2025; Zhu et al., 2023), CGM injects global description context into visual features via gated modulation, yielding stable inputs for downstream reasoning. Since point clouds capture geometric structure while images convey semantic details, CGM uses modality-specific gating networks ($\text{MLP}_p$ and $\text{MLP}_i$) to produce adaptive filters $\boldsymbol{w}_p = \sigma(\text{MLP}_p([\boldsymbol{E}_p, \bar{\boldsymbol{E}}_{d,\text{broad}}]))$ and $\boldsymbol{w}_i = \sigma(\text{MLP}_i([\boldsymbol{E}_i, \bar{\boldsymbol{E}}_{d,\text{broad}}]))$, where $\sigma$ is the sigmoid activation. The grounded representations are computed through element-wise multiplication ($\odot$) with residual connections: $\boldsymbol{E}_p' = \boldsymbol{E}_p + (\boldsymbol{E}_p \odot \boldsymbol{w}_p)$, and $\boldsymbol{E}_i' = \boldsymbol{E}_i + (\boldsymbol{E}_i \odot \boldsymbol{w}_i) \in \mathbb{R}^{N_p \times D_f}$. CGM selectively enhances visual features aligned with the description while preserving the original information through residual connections. The resulting representations $[\boldsymbol{E}_p', \boldsymbol{E}_i']$ serve as primary visual sources for SQA, while ungrounded features $[\boldsymbol{E}_p, \boldsymbol{E}_i]$ are used for VQA. For clarity, we denote these two primary visual sources as $\boldsymbol{E}_1$ and $\boldsymbol{E}_2$ respectively in subsequent sections.

### 3.2 STAGE 2: INFORMATION ATOM EXTRACTION

The core innovation of our framework lies in Question-aware PID (Q-PID), which decomposes the visual sources $\boldsymbol{E}_1$ and $\boldsymbol{E}_2$ into four interpretable information atoms grounded in PID theory: point cloud uniqueness ($U_1$), image uniqueness ($U_2$), redundancy ($R$), and synergy ($S$) using three modules. To make this decomposition question-aware, we employ a cross-attention conditioning mechanism inspired by Alayrac et al. (2022); Tan & Bansal (2019). For any target representation $\boldsymbol{E}_t$, the resulting representation conditioned on question $\bar{\boldsymbol{E}}_{q,\text{broad}}$ is:

$$\boldsymbol{E}_t' = \boldsymbol{E}_t + \text{MHA}(\mathbf{Q} = \boldsymbol{E}_t, \mathbf{K} = \bar{\boldsymbol{E}}_{q,\text{broad}}, \mathbf{V} = \bar{\boldsymbol{E}}_{q,\text{broad}}) \tag{4}$$

To ensure training stability, all extractions employ a residual connection (He et al., 2016).

**Redundancy Extraction** implements Eq. 1, capturing shared information between visual sources. We implement this as a question-agnostic operation since redundancy denotes invariant shared information that remains consistent regardless of query context. To select features likely shared across sources, we compute a per-point, per-channel gate via $\boldsymbol{w}_1 = \sigma(\text{MLP}([\phi_1(\boldsymbol{E}_1), \phi_2(\boldsymbol{E}_2)]))$, where $\phi_1, \phi_2$ are linear projections mapping features into shared representational space for identifying common patterns. Intuitively, $\boldsymbol{w}_1$ serves as a confidence mask indicating which components of $\boldsymbol{E}_1$ are redundant with $\boldsymbol{E}_2$. The redundancy atom is then constructed through selective modulation: $\boldsymbol{R} = \boldsymbol{E}_1 + \text{MLP}(\boldsymbol{w}_1 \odot \phi_1(\boldsymbol{E}_1))$ This approach captures the maximization of Eq. 1 by learning to identify and amplify the most informative shared patterns between modalities, with the MLP enables a non-linear transformation to extract redundancy in a high-dimensional space.

**Uniqueness Extraction** directly implements Eq. 2, capturing information in $\boldsymbol{E}_1$ that predicts the target independently of $\boldsymbol{E}_2$. We achieve this by adaptively filtering off cross modal dependencies.

First, we create a question-aware representation $\boldsymbol{E}_1'$ using Eq. 4, then employ a gating mechanism to identify shared components: $\boldsymbol{w}_2 = \sigma(\text{MLP}([\boldsymbol{E}_1', \boldsymbol{E}_2]))$, where $\boldsymbol{w}_2$ represents the weights of information in $\boldsymbol{E}_1'$ that is predictable from $\boldsymbol{E}_2$. The uniqueness atom is then extracted by filtering off these task-aware dependencies: $\boldsymbol{U}_1 = \boldsymbol{E}_1 + \text{MLP}(\boldsymbol{E}_1' - \boldsymbol{w}_2 \odot \boldsymbol{E}_2)$. This difference approximates the conditional independence operation in Eq. 2, while MLP enables a non-linear extraction. The complement $\boldsymbol{U}_2$ is computed symmetrically.

**Synergy Extraction** conceptually captures emergent information that arises only from the joint consideration of two sources. While Eq. 3 formally defines synergy as the "gap" between joint information and the part attributable to individual sources alone, directly implementing this decomposition in deep models remains non-trivial. In practice, we model synergy using a transformer encoder (Vaswani et al., 2017), $\psi_{\text{Trans}}$, which enables token-level interactions across modalities: $[\boldsymbol{E}_1^{\text{syn}}, \boldsymbol{E}_2^{\text{syn}}] = \psi_{\text{Trans}}([\boldsymbol{E}_1, \boldsymbol{E}_2])$. Here, $\boldsymbol{E}_1^{\text{syn}}$ represents $\boldsymbol{E}_1$ re-contextualized by cross-source interactions. We further condition it on the question (Eq. 4) to obtain $\boldsymbol{E}_1^{\text{syn},\prime}$ and form the synergy atom as $\boldsymbol{S} = \boldsymbol{E}_1 + \boldsymbol{E}_1^{\text{syn},\prime}$. This formulation is a practical approximation of Eq. 3. By modeling synergy as re-contextualization between sources, we retain the emergent information while ensuring robust empirical performance.

### 3.3 STAGE 3: INFORMATION ATOM INTERPRETATION AND REASONING

The final stage dynamically modulates atoms to prioritize relevant information, then fuses them with question embeddings to predict answer $\hat{\alpha}$.

**Dynamic Atom Modulation (DAM)** adaptively weighs different information atoms based on the question while maintaining interpretability. Given decomposed atoms $\{\boldsymbol{U}_1, \boldsymbol{U}_2, \boldsymbol{R}, \boldsymbol{S}\}$, we first pool their features into a global visual summary and concatenate it with the question embedding. A gating network outputs raw logits, which are normalized by temperature-scaled softmax to produce importance weights $\beta = \{\beta_{\boldsymbol{U}_1}, \beta_{\boldsymbol{U}_2}, \beta_{\boldsymbol{R}}, \beta_{\boldsymbol{S}}\} \in \mathbb{R}^4$. To preserve feature magnitudes while enabling adaptive emphasis, we apply multiplicative scaling: $\boldsymbol{A}' \leftarrow (1 + \beta_{\boldsymbol{A}}) \cdot \boldsymbol{A}$, for $\boldsymbol{A} \in \{\boldsymbol{U}_1, \boldsymbol{U}_2, \boldsymbol{R}, \boldsymbol{S}\}$. Concatenating the modulated atoms yields the final visual encoding $\mathbf{E}_v = [\boldsymbol{U}_1', \boldsymbol{U}_2', \boldsymbol{R}', \boldsymbol{S}'] \in \mathbb{R}^{N_p \times 4D_f}$. This design ensures proportional enhancement based on question-specific requirements. The weights $\beta$ provide direct, per-question interpretability of each atom's contribution.

**Cross-Modal Reasoning.** The modulated visual embedding, $\boldsymbol{E}_v$, is then projected to match the question embedding $\boldsymbol{F}_q$, and finally processed by the well-established Multimodal Context-guided Reasoning (MCGR) module from DSPNet (Luo et al., 2025). This module performs iterative fusion between the visual features and the question embeddings, with its final output being fed to a modular co-attention network (Yu et al., 2019) to predict the answer $\hat{\alpha}$.

### 3.4 TRAINING OBJECTIVE

Following Luo et al. (2025)'s work, the main task loss $\mathcal{L}_{task}$ is a linear combination of answer classification, object classification, and reference localization losses. To align learning with PID theory (Eqs. 1–3), we introduce atom-specific **regularization losses** that enforce the theoretical properties of each information component.

**Uniqueness** is enforced by minimizing cosine similarity between each uniqueness atom and the opposite modality: $\mathcal{L}_{\text{unq}} = |\cos(\boldsymbol{U}_1, \boldsymbol{E}_2)| + |\cos(\boldsymbol{U}_2, \boldsymbol{E}_1)|$. This constraint approximates conditional independence by making uniqueness orthogonal, and thus unpredictable, from the other source. **Redundancy** is encouraged through predictive consistency. We use an auxiliary head $\psi_{\text{aux}}$ to ensure the redundancy atom is recoverable from either source: $\mathcal{L}_{\text{red}} = \text{MSE}(\psi_{\text{aux}}(\boldsymbol{R}), \psi_{\text{aux}}(\boldsymbol{E}_1)) + \text{MSE}(\psi_{\text{aux}}(\boldsymbol{R}), \psi_{\text{aux}}(\boldsymbol{E}_2))$. **Synergy** is promoted through an orthogonality constraint, minimizing cosine similarity between the synergy atom and the combination of unique and redundant components: $\mathcal{L}_{\text{syn}} = |\cos(\boldsymbol{S}, \boldsymbol{U}_1 + \boldsymbol{U}_2 + \boldsymbol{R})|$. These surrogates jointly promote redundancy for shared signals, uniqueness for source-specific information, and synergy for emergent interactions.

Beyond PID constraints, we add a **diversity loss** to prevent *atom collapse* by encouraging uniform usage across the batch, where $\bar{\beta}_k$ is the average usage of atom $k$ across batch size $B$, and $K = 4$:

$\mathcal{L}_{\text{div}} = \frac{1}{K} \sum_{k \in \{\mathbf{U}_1, \mathbf{U}_2, \mathbf{R}, \mathbf{S}\}} \left( K \cdot \bar{\beta}_k - 1 \right)^2$. Therefore, the final objective becomes, where $\lambda_u = 1.0$, $\lambda_r = \lambda_s = 0.1$, and $\lambda_d = 0.01$: $\mathcal{L}_{3DQA} = \mathcal{L}_{\text{task}} + \lambda_u \mathcal{L}_{\text{unq}} + \lambda_r \mathcal{L}_{\text{red}} + \lambda_s \mathcal{L}_{\text{syn}} + \lambda_d \mathcal{L}_{\text{div}}$ .

## 4 EXPERIMENT

We validate ATOM on two 3D question answering benchmarks: ScanQA (Azuma et al., 2022) for 3D visual question answering and SQA3D (Ma et al., 2022) for 3D situated question answering. On the ScanQA dataset, we evaluate using standard accuracy metrics (EM@1, EM@10) and sentence-level metrics (BLEU-4, ROUGE-L, METEOR, CIDEr). On the SQA3D dataset, we adopt the answer accuracy. Full detail in App. C.2.

### 4.1 RESULTS ON SCANQA DATASET

Table 1: Answer accuracy on ScanQA test set. Each entry denotes "test w/ object" / "test w/o object". The best results are marked **bold**, and the second-best ones are underlined.

| Method | Pre-trained | EM@1 | EM@10 | BLEU-4 | ROUGE | METEOR | CIDEr |
|---|---|---|---|---|---|---|---|
| Image+MCAN (Azuma et al., 2022) | × | 22.3 / 20.8 | 53.1 / 51.2 | 14.3 / 9.7 | 31.3 / 29.2 | 12.1 / 11.5 | 60.4 / 55.6 |
| ScanRefer+MCAN (Azuma et al., 2022) | × | 20.6 / 19.0 | 52.4 / 49.7 | 7.5 / 7.8 | 30.7 / 28.6 | 12.0 / 11.4 | 57.4 / 53.4 |
| ScanQA (Azuma et al., 2022) | × | 23.5 / 20.9 | 56.5 / 54.1 | 12.0 / 10.8 | 34.3 / 31.1 | 13.6 / 12.6 | 67.3 / 60.2 |
| Multi-CLIP (Delitzas et al., 2023) | ✓ | 24.0 / 21.5 | - / - | 12.7 / 12.9 | 35.4 / 32.6 | 14.0 / 13.4 | 68.7 / 63.2 |
| 3D-VisTA (Zhu et al., 2023) | × | 25.2 / 20.4 | 55.2 / 51.5 | 10.5 / 8.7 | 35.5 / 29.6 | 13.8 / 11.6 | 68.6 / 55.7 |
| 3D-VisTA (Zhu et al., 2023) | ✓ | **27.0** / 23.0 | 57.9 / 53.5 | **16.0** / 11.9 | 38.6 / 32.8 | 15.2 / 12.9 | 76.6 / 62.6 |
| 3DGraphQA (Wu et al., 2024) | × | 25.6 / 22.3 | 58.7 / **56.1** | 15.1 / 12.9 | 36.9 / 33.0 | 14.7 / 13.6 | 74.6 / 62.9 |
| DSPNet (Luo et al., 2025) | × | 26.5 / **23.8** | 58.8 / **56.1** | 15.4 / **15.7** | **39.3** / **35.1** | **15.7** / 14.3 | **78.1** / **69.6** |
| **ATOM (ours)** | × | 26.7 / 22.8 | **58.9** / 55.8 | 15.0 / 15.6 | 38.7 / 33.5 | 15.5 / 14.0 | 77.3 / 67.5 |

**Baseline.** We compare against several baselines on ScanQA, including 2D image VQA approaches such as Image+MCAN and ScanRefer+MCAN (Yu et al., 2019; Azuma et al., 2022), 3D-specific methods including ScanQA (Azuma et al., 2022) and 3DGraphQA (Wu et al., 2024), pre-trained models Multi-CLIP (Delitzas et al., 2023) and 3D-VisTA (Zhu et al., 2023), and DSPNet (Luo et al., 2025), which represents the current SOTA on this dataset.

**Results Analysis.** In Tab. 1, ATOM demonstrates competitive performance across evaluation metrics while achieving interpretable 3D reasoning. Our method ranks among the top performers on EM@1, indicating strong spatial understanding capabilities. Notably, ATOM achieves the highest performance on EM@10, demonstrating superior answer diversity and robustness in top-K predictions. Particularly, the strong scores across sentence-level metrics (BLEU-4, ROUGE, METEOR) reflect effective semantic alignment with reference answers and comprehensive coverage of key information. While our CIDEr performance is slightly below the peak baseline, this represents the expected trade-off between maintaining interpretable information decomposition and optimizing pure task metrics. The consistent performance across both object-present and object-absent test conditions validates our framework's robustness across diverse question types and scene complexities. Tab. 3 confirms ATOM significantly outperforms DSPNet in the multi-seed experiments (23.57±0.18% vs 22.98±0.05%, p<0.01) on ScanQA validation set. Crucially, ATOM achieves these competitive results without requiring pre-training, operating entirely through end-to-end learning while providing transparent reasoning pathways through Q-PID decomposition and DAM.

### 4.2 RESULTS ON SQA3D DATASET

**Baseline.** We evaluate ATOM against diverse baselines on SQA3D, including egocentric video models ClipBERT (Lei et al., 2021) and MCAN (Yu et al., 2019), 3D QA methods: ScanQA (Azuma et al., 2022) and SQA3D (Ma et al., 2022), pre-trained models: Multi-CLIP (Delitzas et al., 2023) and 3D-VisTA (Zhu et al., 2023), graph-based method: 3DGraphQA (Wu et al., 2024), and current SOTA DSPNet (Luo et al., 2025).

**Results Analysis.** ATOM achieves 49.7 EM@1 average performance on SQA3D, ranking second across all baselines (Tab. 2). Notably, the performance distribution reveals distinct advantages of our information-theoretic approach. ATOM dominates *Is* questions while maintaining competitive

Table 2: The question answering accuracy on the SQA3D dataset. In the test set column, the brackets indicate the number of samples for each type of question. The best results are in **bold**, and the second-best ones are underlined.

| Method | Pre-trained | What (1,147) | Is (652) | How (465) | Can (338) | Which (351) | Other (566) | EM@1 |
|---|---|---|---|---|---|---|---|---|
| ClipBERT (Lei et al., 2021) | × | 30.2 | 60.1 | 38.7 | 63.3 | 42.5 | 42.7 | 43.3 |
| MCAN (Yu et al., 2019) | × | 28.9 | 59.7 | 44.1 | 68.3 | 40.7 | 40.5 | 43.4 |
| ScanQA (Azuma et al., 2022) | × | 28.6 | 65.0 | 47.3 | 66.3 | 43.9 | 42.9 | 45.3 |
| SQA3D (Ma et al., 2022) | × | 33.5 | 66.1 | 42.4 | 69.5 | 43.0 | 46.4 | 47.2 |
| Multi-CLIP (Delitzas et al., 2023) | ✓ | - | - | - | - | - | - | 48.0 |
| 3D-VisTA (Zhu et al., 2023) | × | 32.1 | 62.9 | 47.7 | 60.7 | 45.9 | 48.9 | 46.7 |
| 3D-VisTA (Zhu et al., 2023) | ✓ | 34.8 | 63.3 | 45.4 | **69.8** | 47.2 | 48.1 | 48.5 |
| 3DGraphQA (Wu et al., 2024) | × | 36.4 | 64.7 | 46.1 | **69.8** | **47.6** | 48.2 | 49.2 |
| DSPNet (Luo et al., 2025) | × | **38.2** | 66.0 | **51.2** | 66.6 | 42.5 | **51.6** | **50.4** |
| ATOM (ours) | × | 37.6 | **66.5** | 48.7 | 68.5 | 45.7 | 51.1 | 49.7 |

performance on challenging *What*, *How*, and *Other* questions. This performance pattern demonstrates that our information-theoretic approach provides particular advantages for verification tasks while remaining robust across diverse reasoning demands. Compared to other top-performing methods, ATOM exhibits more balanced competency: while DSPNet excels on specific categories (*What*, *How*, *Other*) and 3DGraphQA dominates spatial reasoning questions (*Can*, *Which*), ATOM maintains consistently high performance across the full spectrum of question types. This balanced effectiveness suggests that Q-PID information decomposition offers fundamental advantages over conventional multimodal fusion techniques. Tab. 3 demonstrates that ATOM achieves statistically similar performance to DSPNet (p=0.098), despite a slightly lower mean EM@1% (49.54±0.11% vs. 49.80±0.08%). This demonstrates ATOM maintains competitive accuracy while uniquely providing full interpretability through quantifiable information atoms, enabling both effective and transparent 3D reasoning.

Table 3: Multi-seed validation ($N$=4). *p<0.05, **p<0.01.

| Dataset | DSPNet | ATOM | $\Delta$ | p |
|---|---|---|---|---|
| ScanQA | 22.98±0.05 | **23.57±0.18** | +0.59 | 0.007** |
| SQA3D | 49.80±0.08 | 49.54±0.11 | -0.26 | 0.098 |

Table 4: Ablation study on essential components. Results show EM@1 on ScanQA validation and SQA3D test datasets. "–" denotes not applicable. Complete ablations in App. D.1.

| Dataset | TextConcat | w/o CGM | w/o Q-PID | $U_1$ | $U_2$ | $U_1 + U_2$ | $R$ | $S$ | $R + S$ | ATOM |
|---|---|---|---|---|---|---|---|---|---|---|
| ScanQA | – | – | 22.50 | 22.57 | 22.61 | 22.86 | 22.62 | 22.65 | 22.74 | **23.52** |
| SQA3D | 49.42 | 49.05 | 48.76 | 48.83 | 48.86 | 48.98 | 48.91 | 49.02 | 49.37 | **49.71** |

## 4.3 ABLATION STUDY

We validate ATOM's essential components through systematic ablations (Tab. 4). Complete results are in App. D.1. Using concatenated question-description encoding (TextConcat) reduces SQA3D performance to 49.42%, confirming that distinguishing grounding from reasoning is necessary. Eliminating description conditioning (w/o CGM) reduces SQA3D to 49.05%, validating that grounding visual context is essential. Most critically, replacing Q-PID with naive concatenation (w/o Q-PID) degrades performance to 22.50%/48.76%, demonstrating the fundamental value of information-theoretic disentanglement. We systematically examine individual atom contributions. Individual uniqueness atoms ($U_1$, $U_2$) and interaction atoms ($R$, $S$) each provide modest improvements over w/o Q-PID baseline. Combining uniqueness atoms ($U_1 + U_2$) yields +0.36%/+0.22% gains, while combining interaction atoms ($R+S$) achieves stronger +0.24%/+0.61% improvements, demonstrating complementarity within each atom category. Crucially, full ATOM incorporating all four atoms provides substantial gains of +0.78%/+0.34% over $R + S$ and +0.66%/+0.73% over $U_1 + U_2$. This validates that uniqueness and interaction atoms capture distinct, complementary information patterns, both systematically overlooked by conventional fusion, and their principled combination is essential for 3D QA.

## 4.4 QUALITATIVE RESULTS

Fig. 4 (**left**) illustrates DAM weight distributions across question types on SQA3D and ScanQA datasets. Uniqueness and synergy atoms remain essential across all question types in both datasets,

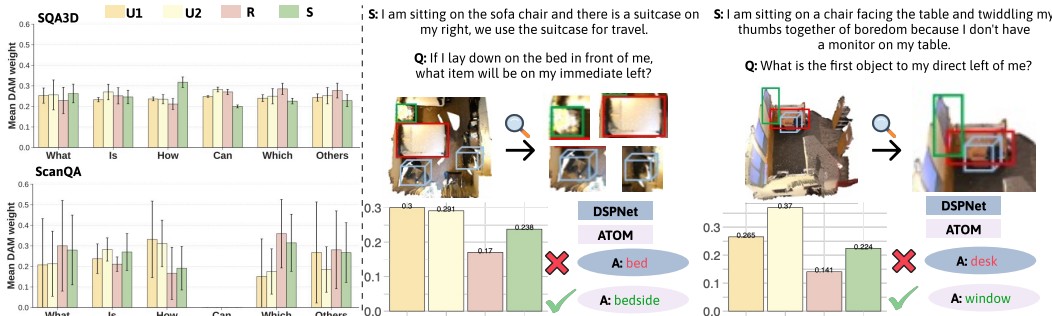

Figure 4: **Left**: Mean DAM weight distributions by question type on SQA3D and ScanQA datasets. ATOM adaptively emphasizes different information atoms (U1: Point Cloud Uniqueness, U2: Image Uniqueness, R: Redundancy, S: Synergy). Note that ScanQA lacks *Can* questions due to its non-situated nature. Error bars indicate standard deviation. **Right**: Qualitative comparisons showing agent positions (blue cubes), DSPNet predictions (red boxes), & ATOM predictions (green boxes). Sample-level DAM weights demonstrate ATOM's adaptive information prioritization.

confirming their fundamental role in 3D QA. ScanQA exhibits high redundancy and synergy across questions, with a notable decline for *How* questions, where reasoning relies more on uniqueness in the absence of context. In contrast, SQA3D demonstrates flatter distributions as situated descriptions provide explicit grounding that distributes reasoning load evenly across atoms. Fig. 4 (**right**) demonstrates ATOM's enhanced reasoning in complex scenarios. DSPNet struggles with correct object identification in both cases, while ATOM successfully localizes target objects. These examples highlight two critical limitations of existing approaches: (1) difficulty maintaining object-question alignment when descriptions involve multiple objects, and (2) challenges filtering irrelevant information when descriptions reference objects unrelated to the question target. ATOM's success validates the importance of CGM, which explicitly separates description processing from question reasoning and injects contextual knowledge into visual features. The sample-level DAM weight distributions reveal that *What* questions rely heavily on uniqueness information, as models must identify distinguishing object characteristics (shape, appearance) that enable precise localization. Meanwhile, synergy captures inter-object spatial relationships crucial for 3D reasoning.

### 4.5 USER STUDY

We validate ATOM's interpretability through a user study with five participants (App. F).

**Procedure**  The study was conducted remotely via Zoom and lasted approximately one hour. Prior to the session, participants received an information sheet detailing the study's purpose and instructions, then completed a consent form and demographic questionnaire. We began by explaining the definitions of our four information atoms (point cloud uniqueness, image uniqueness, redundancy, and synergy) using example 3D point clouds and multiview images from both ScanQA and SQA3D datasets. Participants then completed six warm-up questions to familiarize themselves with the task format. During the warm-up session, for each question, participants were presented with: (1) the question and context, (2) the ground-truth answer, (3) a manipulable 3D point cloud visualization, (4) uniformly sampled multiview images as used during model inference, and (5) two information atom weight distributions: one generated by ATOM and one by GPT5 (OpenAI, 2025). Participants were asked to reason about their answer derivation process and select which distribution better aligned with their reasoning. The main study comprised 20 questions (*4 How, 6 What, 2 Can, 2 Which, 5 Where, 1 Other*) evenly split between ScanQA and SQA3D, presented in alternating order. For each question, participants ranked the four information atoms in order of importance for arriving at the answer. To ensure they did not randomly select an atom, we asked them to justify their decisions. A five-minute break was provided after the first 10 questions.

**Results**  ATOM demonstrates strong alignment with human expert reasoning patterns, with marked dataset-specific characteristics. For SQA3D, Cohen's kappa analysis reveals substantial-to-almost-perfect individual agreement ($\mu = 0.76, \sigma = 0.12$), with three out of five participants achieving $\kappa > 0.80$, indicating that ATOM's information atom rankings closely mirror expert spatial reason-

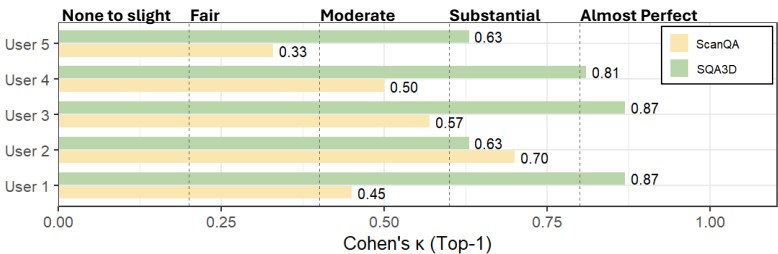

Figure 5: Cohen's kappa ($\kappa$) between human judgments and ATOM's top-1 atom predictions.

ing strategies for situated question answering. For ScanQA, individual agreement remained in the fair-to-substantial range ($\mu = 0.51, \sigma = 0.13$), reflecting meaningful but more variable alignment across participants. The cross-dataset consistency within individuals is noteworthy: Users 1, 3, and 4 demonstrated high agreement with ATOM across both datasets (SQA3D: $\kappa = 0.81\text{-}0.87$; ScanQA: $\kappa = 0.45\text{-}0.57$), while Users 2 and 5 exhibited more variable patterns (SQA3D: $\kappa = 0.63\text{-}0.70$; ScanQA: $\kappa = 0.33\text{-}0.70$). Critically, no participant exhibited poor agreement ($\kappa < 0.20$) with ATOM in either dataset, establishing that ATOM maintains meaningful alignment across diverse expert reasoning strategies. The consistently higher kappa values in SQA3D compared to ScanQA across all participants suggests that ATOM aligns more reliably with human cognition when spatial reasoning is explicitly grounded in situated descriptions, whereas free-form scene understanding admits greater strategic diversity in atom prioritization. Participants reported that ATOM enables detection of *spurious correctness*, a capability black-box models lack, where models predict correct answers despite using insufficient modalities. ATOM explicitly reveals which modality receives genuine attention versus which contains task-relevant information, exposing cases like correct predictions despite absent spatial relationships in images.

### 4.6 DISCUSSION

Our DAM weight analysis (Fig. 4 and App. G) reveals that different question types require distinct information atom patterns, validating ATOM's information-theoretic foundation. *What* questions rely heavily on uniqueness atoms for precise object identification, *How* questions predominantly utilize synergy for spatial relationship reasoning, *Can* queries depend on redundancy for cross-modal confirmation, while *Is* questions exhibit balanced distributions across all atoms. Interestingly, these patterns align with human spatial reasoning: humans use distinctive features for identification (uniqueness), integrate multiple cues for spatial understanding (synergy), and seek cross-modal confirmation for navigation decisions (redundancy). A more comprehensive analysis is provided in App. E. This correspondence suggests ATOM's decomposition captures fundamental principles of multimodal spatial cognition, providing a more interpretable and robust approach to 3D reasoning than conventional black-box fusion methods. These findings have implications for future multimodal AI systems. The alignment between ATOM's information patterns and human spatial cognition suggests that information-theoretic decomposition could serve as a foundational principle for developing more interpretable and robust reasoning architectures. Furthermore, the question-specific utilization patterns provide a framework for designing adaptive systems that emphasize appropriate information types based on task requirements, potentially extending beyond 3D reasoning to other multimodal domains requiring explainable AI performance. However, ATOM is currently limited to closed-set QA with predefined answer vocabularies, constraining its applicability to open-ended reasoning scenarios (full details in App. B).

### 5 CONCLUSION

We present ATOM, the first information-theoretic framework that operationalizes Partial Information Decomposition for interpretable 3D QA. ATOM extracts and grounds visual features via Query-drive View Aggregation and Contextual Grounding Module components. It decomposes point cloud and image features into four explainable information atoms using a Question-aware PID module. Finally, a Dynamic Atom Modulation module is introduced to prioritize information atoms upon the question. Experimental results demonstrate that ATOM achieves competitive performance while providing unprecedented interpretability, establishing a new paradigm for transparent 3D reasoning and opening promising directions for interpretable embodied AI.

## 6 ETHICS STATEMENT

We acknowledge and adhere to the ICLR Code of Ethics. This work advances interpretable AI for 3D spatial understanding, enhancing trustworthiness of embodied AI systems in robotics and autonomous applications. We uphold scientific excellence through rigorous validation, transparent reporting, and reproducible code. Our research uses publicly available datasets (ScanQA and SQA3D based on ScanNet) collected following appropriate ethical guidelines with proper consent and licensing. We respect intellectual property of all referenced works and provide appropriate attribution. The user study was approved by our university's Institutional Review Board (IRB). All participants provided written informed consent. The IRB approval number will be disclosed upon acceptance to maintain review anonymity. All data was collected, stored, and analyzed following ethical research standards and privacy regulations. The ATOM framework does not involve other human subjects research, harmful applications, or raise concerns regarding discrimination, bias, privacy violations, or legal compliance. We declare no conflicts of interest.

## 7 REPRODUCIBILITY STATEMENT

We have made comprehensive efforts to ensure the reproducibility of our work. Complete implementation details, including network architectures, hyperparameters, and training procedures, are provided in the Appendix. Anonymous source code with detailed documentation is included in the supplementary materials as a zipped file, containing scripts for data preprocessing, model training, and evaluation protocols for both ScanQA and SQA3D datasets. All experimental configurations, loss function implementations, and evaluation metrics are fully specified in the main text and the supplementary materials. The datasets used (ScanQA and SQA3D) are publicly available from their official sources, with data processing steps documented in our code repository.

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

## A  USE OF LLMS

Large Language Models (LLMs) were used solely to aid and polish the writing of this paper. All technical contributions, experimental design, implementation, and analysis are entirely the work of the authors.

## B  LIMITATIONS

ATOM has several limitations that present opportunities for future work. First, our framework is constrained to closed-set 3D QA with predefined answer vocabularies. Future work could explore open-set scenarios by integrating pre-trained vision-language models or leveraging large-scale scene-text paired datasets to enable free-form answer generation. Second, while we demonstrate PID operationalization for two modalities (point clouds and images), extending to multiple modalities ($>2$) remains challenging due to the exponential growth of interaction terms, requiring novel approaches to balance interpretability with computational efficiency. Additionally, the explicit information decomposition requires careful hyperparameter tuning.

## C  IMPLEMENTATION DETAILS

### C.1  MODAL ENCODER ARCHITECTURE

We provide finer details on the architecture and rationale for the modality-specific encoders used in our framework.

**3D Encoder.** We process the input RGB-colored point cloud $\boldsymbol{P} \in \mathbb{R}^{N \times 6}$ using a pre-trained PointNet++ (Qi et al., 2017) encoder, with the weights initialized from VoteNet (Qi et al., 2019). We specifically choose a point-level encoder over an object-centric one to ensure our model has access to a comprehensive scene representation. Object detectors can overlook non-object regions (e.g., floors, walls) and abstract away fine-grained geometric details that may be crucial for answering certain questions.

PointNet++(Qi et al., 2017) is a hierarchical neural network that processes a point set in a metric space. Its architecture consists of a series of set abstraction layers, which iteratively subsample and group points to learn features at increasingly larger spatial scales. This allows it to capture both local geometric details and broader contextual information. The encoder outputs a set of $N_p$ seed point features, yielding a raw feature matrix $\boldsymbol{F}_p^{\text{raw}} \in \mathbb{R}^{N_p \times D_p}$, where each feature vector corresponds to a specific location in the 3D scene.

**Image Encoder.** For the $M$ multi-view images, we employ a frozen Swin Transformer (Liu et al., 2021) as a feature extractor. The Swin Transformer is a hierarchical Vision Transformer (Dosovitskiy et al., 2020) that has demonstrated state-of-the-art performance on a range of vision tasks. Unlike earlier transformer architectures (Vaswani et al., 2017) that compute global self-attention, it introduces a shifted window mechanism, which limits self-attention computation to non-overlapping local windows that are shifted across layers. This approach is not only more computationally efficient but also allows for cross-window connections, enabling it to model dependencies at multiple scales effectively.

By keeping the encoder frozen, we leverage the powerful and generalizable visual concepts learned during its pre-training on large-scale datasets like ImageNet (Deng et al., 2009). From its final layer,

we extract a set of feature maps $\boldsymbol{F}_i^{\text{raw}} \in \mathbb{R}^{M \times H \times W \times D_i}$, where $H \times W$ is the spatial resolution of the feature grid and $D_i$ is the feature dimension.

**Text Encoder.** Language inputs, the question $\boldsymbol{q}$ and (for SQA) the description $\boldsymbol{d}$, are processed by a shared Sentence-BERT (SBERT) (Reimers & Gurevych, 2019) model. SBERT is a modification of the BERT architecture specifically fine-tuned to produce semantically meaningful sentence embeddings. It uses a Siamese and Triplet network structure to yield embeddings where sentences with similar meanings are close in the vector space, making it highly suitable for capturing the semantic intent of questions and descriptions. We use a single, shared transformer for both inputs to ensure consistency and parameter efficiency. The encoder produces token-level embeddings $\boldsymbol{F}_q \in \mathbb{R}^{L_q \times D_t}$ and $\boldsymbol{F}_d \in \mathbb{R}^{L_d \times D_t}$, where $L$ and $D_t$ denote sequence length and embedding dimension respectively.

**Unified Projection.** To enable meaningful cross-modal operations in the subsequent modules, these initial features are mapped into a unified embedding space of dimension $D_f$. For point cloud and language features, this projection is applied directly via learnable linear layers:

$$\boldsymbol{E}_p = \phi_p(\boldsymbol{F}_p^{\text{raw}}) \in \mathbb{R}^{N_p \times D_f} \quad \bar{\boldsymbol{E}}_q = \phi_q(\text{Pool}(\boldsymbol{F}_q)) \quad \text{and} \quad \bar{\boldsymbol{E}}_d = \phi_d(\text{Pool}(\boldsymbol{F}_d)) \in \mathbb{R}^{D_f} \quad (5)$$

where $\phi_p$, $\phi_q$, and $\phi_d$ are the modality-specific projection functions, and $\text{Pool}(\cdot)$ aggregates the token embeddings into a single global vector (e.g., by averaging). Projections for the image features ($\boldsymbol{F}_i^{\text{raw}}$) are applied dynamically within the downstream modules to allow for more context-specific transformations.

## C.2 EXPERIMENT CONFIGURATION

**Datasets.** The ScanQA dataset (Azuma et al., 2022) contains 41,363 questions and 58,191 answers. The training and validation splits followed ScanRefer (Chen et al., 2020), while ScanQA includes two test sets with and without object annotations. The SQA3D (Ma et al., 2022) consists of around 21k descriptions and 35k questions base on 650 ScanNet (Dai et al., 2017) scenes.

**Architectural Details.** Our framework builds upon DSPNet (Luo et al., 2025), inheriting its data preprocessing pipeline. For multi-view images, we uniformly sample 20 views per scene at 224×224 resolution. Additionally, for each scene, we sample 40,000 points from raw point clouds, using random sampling during training and farthest point sampling (FPS) at inference. We employ pre-trained encoders: a PointNet++ from VoteNet (Qi et al., 2019), a frozen Swin Transformer (Liu et al., 2021) for images, and an MPNet-based SBERT (Reimers & Gurevych, 2019) for language. We follow Luo et al. (2025)'s MCGR module settings, using $K = 256$ in the FPS stage with hidden dimension 768. Our ATOM-specific configurations include a unified feature dimension of $D_f = 512$, and a shared hidden dimension of 2048 for all PID-related modules, including the conditioner $\psi_{\text{cond}}$, the transformer encoder $\psi_{\text{Trans}}$, and the gating networks $\phi_{\text{gate}}$. The temperature for the softmax function within DAM is set to 1.5.

**Training and Optimization.** We train our model for 12 epochs using the AdamW (Loshchilov & Hutter, 2017) optimizer ($\beta_1 = 0.9, \beta_2 = 0.999$, weight decay $1 \times 10^{-5}$). Training is performed on 4 NVIDIA RTX 4090 GPUs with a per-GPU batch size of 12. We use a learning rate schedule with a 500-step warmup from $5 \times 10^{-5}$ to $1 \times 10^{-4}$, followed by a cosine decay back to $5 \times 10^{-5}$. The language encoder uses a learning rate scaled by a factor of 0.1. The PID regularization coefficients in our final loss are set to $\lambda_u = 1.0$, $\lambda_r = \lambda_s = 0.1$ and $\lambda_d = 0.01$, while the main task loss is given a weight of 1 according to Luo et al. (2025).

## D ABLATION STUDY

## D.1 COMPLETE ABLATION STUDY

We present comprehensive ablations examining model designs and information atoms on ScanQA validation and SQA3D test splits. Tab. 5(a) evaluates architectural components, while Tab. 5(b) provides exhaustive atom combination analysis.

**Model Design Components.** TextConcat concatenates question and description without explicit separation, reducing SQA3D performance to 49.42%, confirming that distinguishing grounding from reasoning is necessary for situated QA. Removing question-view attention (w/o QVA) causes significant drops to 22.40%/49.64%, highlighting the importance of joint spatial-semantic selection for effective multi-view aggregation. Eliminating description conditioning from visual features (w/o CGM) reduces SQA3D to 49.05%, validating that grounding visual context in situational descriptions is essential. Replacing PID decomposition with naive concatenation (w/o Q-PID) substantially degrades performance to 22.50%/48.76%, demonstrating the fundamental value of information-theoretic disentanglement over conventional fusion. Removing dynamic atom modulation (w/o DAM) leads to clear drops (23.17%/49.51%), proving that question-adaptive atom prioritization is vital for handling diverse reasoning patterns. Finally, eliminating PID regularization (w/o reg) reduces performance by 0.25%/0.72%, indicating that theoretical constraints are essential for stable atom decomposition and generalization.

**Information Atoms.** Tab. 5(b) systematically examines all meaningful atom combinations, where "$\times$" indicates masking with zero tensors. The all-masked baseline ($\times/\times/\times/\times$: 22.50%/48.76%) is equivalent to w/o Q-PID, representing naive concatenation. Individual atoms provide modest but consistent gains: point cloud uniqueness ($U_1$: 22.57%/48.83%), image uniqueness ($U_2$: 22.61%/48.86%), redundancy ($R$: 22.62%/48.91%), and synergy ($S$: 22.65%/49.02%) each outperform the baseline, confirming that each atom captures valuable information patterns.

Combining atoms within categories reveals complementarity: uniqueness combination ($U_1 + U_2$: 22.68%/48.98%) yields +0.18%/+0.22% improvement over baseline, while interaction combination ($R + S$: 22.74%/49.37%) achieves stronger +0.24%/+0.61% gains. Cross-category combinations exhibit dataset-dependent patterns: $U_1 + U_2 + R$ (22.40%/49.59%) excels on SQA3D (+0.83%) but underperforms on ScanQA (-0.10%), suggesting synergy is particularly critical for open-ended scene understanding in ScanQA. Conversely, $U_1 + U_2 + S$ (23.14%/48.65%) shows strong ScanQA performance (+0.64%) but weaker SQA3D results (-0.11%), indicating redundancy plays a more important role in situated reasoning with explicit spatial grounding.

Most importantly, full ATOM incorporating all four atoms achieves optimal performance (23.52%/49.71%), substantially outperforming all partial combinations: +1.02%/+0.95% over baseline, +0.78%/+0.34% over $R + S$, and +0.84%/+0.73% over $U_1 + U_2$. This validates that all four PID atoms—point cloud uniqueness, image uniqueness, redundancy, and synergy—capture distinct, complementary information patterns essential for comprehensive 3D QA reasoning, and their principled combination through Q-PID decomposition is crucial for achieving optimal performance.

Table 5: Complete ablation study on model designs (**a**) and information atoms (**b**). Results show EM@1 on ScanQA validation and SQA3D test datasets. "–" denotes not applicable. "$\times$" indicates masking atoms with zero tensors.

(a) Model Design Components

| Dataset | TextConcat | w/o QVA | w/o CGM | w/o Q-PID | w/o DAM | w/o reg | ATOM |
|---------|-----------|---------|---------|-----------|---------|---------|------|
| ScanQA | – | 22.40 | – | 22.50 | 23.17 | 23.27 | **23.52** |
| SQA3D | 49.42 | 49.64 | 49.05 | 48.76 | 49.51 | 48.99 | **49.71** |

(b) Information Atoms ($U_1/U_2/R/S$). "$\times$" indicates masking atoms with zero tensors.

| Atoms | $\times/\times/\times/\times$ | $U_1$ | $U_2$ | $U_1 + U_2$ | $R$ | $S$ | $U_1 + U_2 + R$ | $U_1 + U_2 + S$ | $R + S$ | ATOM |
|-------|------|-------|-------|-------------|-----|-----|-----------------|-----------------|---------|------|
| ScanQA | 22.50 | 22.57 | 22.61 | 22.68 | 22.62 | 22.65 | 22.40 | 23.14 | 22.74 | **23.52** |
| SQA3D | 48.76 | 48.83 | 48.86 | 48.98 | 48.91 | 49.02 | 49.59 | 48.65 | 49.37 | **49.71** |

## D.2 ARCHITECTURAL DESIGN

We further compare our ATOM with two variants: **ATOM-MoE**[1], which strictly follows the I²MoE implementation (Xin et al., 2025) but adapts to 3D QA, and **ATOM-Flat**, which treats the SQA description as an independent third modality.

---

[1]Since I²MoE (Xin et al., 2025) is not open-sourced, we implement this baseline following their architectural specifications.

**ATOM-MoE**    replaces our core Question-aware PID (Sec. 3.2) and Dynamic Atom Modulation (Sec. 3.3) modules with the Mixture-of-Experts (MoE) architecture proposed by I$^2$MoE (Xin et al., 2025). This variant instantiates four distinct **interaction experts** to separately model uniqueness for point clouds $U_1$, uniqueness for multi-view images $U_2$, redundancy $R$, and synergy $S$. Following the I$^2$MoE design, each expert is composed of a lightweight MCGR module (Luo et al., 2025) and a dedicated prediction head (Yu et al., 2019). All four experts receive the full visual features $(E_p, E_i)$ and the question embedding $(F_q)$ as input, each generating a separate answer prediction. A separate **gating network**, implemented as an MLP, also takes the global visual features to produce four importance weights, one for each expert. The final prediction is a weighted sum of the outputs from the four expert heads. Crucially, to enforce expert specialization, we adopt the **interaction loss** from I$^2$MoE, which uses forward passes with **masked inputs** as a form of weak supervision to train each expert for its designated interaction type.

**ATOM-Flat**    discards the Contextual Grounding Module (Sec. 3.1) entirely. It instead treats the textual description as a distinct third input modality ($E_d$) alongside the point cloud features ($E_p$) and multi-view image features ($E_i$). To be consistent with our work (Sec. 3.2), we denote these embeddings as $E_1 = E_p$, $E_2 = E_i$, and $E_3 = E_d$.

Consequently, this variant extends our Q-PID framework (see Fig. 2) to the trivariate case. It decomposes the three modalities into eight distinct information atoms, as illustrated in Fig. 6. These eight atoms consist of unique contributions from each modality, pairwise redundancies, a triple redundancy, and the overall synergy. Following this decomposition, the eight atoms are refined by the DAM module (Sec. 3.3) and processed by the MCGR module (Luo et al., 2025) to predict the final answer $\hat{\alpha}$, which follows the same downstream pipeline as our main model. This design allows us to directly compare the efficacy of structured, hierarchical grounding (original ATOM) against a 'flat', fully decomposed multimodal fusion approach.

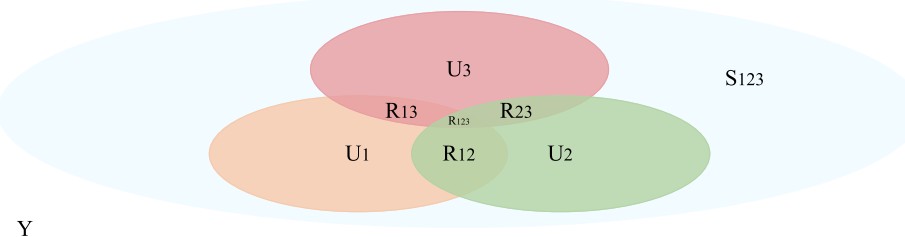

Figure 6: Partial Information Decomposition (PID) of the total information $I(\{X_1, X_2, X_3\}; Y)$ into eight atoms: unique contributions from each modality ($U_1, U_2, U_3$), pairwise redundancies ($R_{12}, R_{13}, R_{23}$), triple redundancy ($R_{123}$), and the synergistic information that emerges only when all three modalities are considered jointly ($S_{123}$).

**Results.**    The results presented in Tab. 6 validate our architectural choices. On SQA3D, **ATOM-Flat** shows a notable degradation (48.01 vs. **49.71**) due to loss of contextual grounding and increased complexity from trivariate decomposition (8 atoms vs. 4 atoms). This drop suggests that our hierarchical approach—using the textual description to explicitly ground visual features—is significantly more effective than treating the description as a flat, independent modality. **ATOM-MoE** underperforms our full **ATOM** on both datasets (22.80 vs. **23.52** on ScanQA; 48.52 vs. **49.71** on SQA3D). While its competent performance confirms the general benefit of disentangling information, its inferiority to ATOM underscores the advantages of our proposed method. We attribute this to two factors: 1) our Q-PID module's direct use of question semantics to guide the decomposition provides a more powerful and task-aware conditioning signal than the indirect supervision from input masking used in MoE-style losses, and 2) our DAM module allows for explicit, fine-grained modulation of the final information atoms before reasoning. Collectively, these ablations confirm that ATOM's specific combination of structured grounding and question-aware decomposition is superior to these alternative designs.

Table 6: Ablation on architectural design comparing ATOM-MoE which follows I²MoE (Xin et al., 2025) implementation, and ATOM-Flat which treats the description as a third modality, with our **ATOM** framework on the validation split of ScanQA dataset and test split of SQA3D dataset (reporting EM@1). "–" denotes results not applicable.

| Methods | ScanQA | SQA3D |
|---|---|---|
| ATOM-MoE | 22.80 | 48.52 |
| ATOM-Flat | - | 48.01 |
| **ATOM (ours)** | **23.52** | **49.71** |

# E   MORE QUANTITATIVE RESULTS

## E.1   COMPREHENSIVE ANALYSIS ON INFORMATION ATOMS

To empirically validate the adaptive behavior of our Dynamic Atom Modulation (DAM) mechanism and address the fundamental question of how information atoms manifest across different grounding contexts, we conducted comprehensive multi-seed analysis across four random initializations (seeds: 42, 1024, 2021, 3407). We analyzed 31,744 samples in total, extracting the learned importance weights $\{\beta_{U_1}, \beta_{U_2}, \beta_R, \beta_S\}$ for each inference sample. This analysis reveals a critical interpretability principle: *situated grounding enables cleaner decomposition of cognitive reasoning processes, while its absence triggers compensatory adaptation strategies*. We demonstrate that these dataset-specific patterns, rather than contradicting our framework, validate its capacity to expose how models adaptively reconfigure information integration under varying input conditions.

**Dataset-Level Distributional Characteristics.** Tab. 7 presents per-atom weight statistics aggregated across all question types for each dataset, revealing fundamental differences in decomposition behavior. SQA3D exhibits remarkably tight distributions for each individual atom (standard deviation $\sigma \approx 0.04$–$0.09$ for $U_1$, $U_2$, $R$, and $S$), with relatively balanced mean weights ($\beta_{U_1} = 0.237$, $\beta_{U_2} = 0.252$, $\beta_R = 0.245$, $\beta_S = 0.266$). This statistical signature indicates that situated descriptions provide consistent grounding context, enabling stable, cognitively-aligned decomposition patterns. In contrast, ScanQA demonstrates markedly wider per-atom distributions ($\sigma \approx 0.14$–$0.23$), with elevated redundancy emphasis ($\beta_R = 0.295$ vs. SQA3D's $0.245$). This distributional broadening reflects three interrelated factors: (1) absence of situational anchoring introduces inherent query ambiguity, (2) diverse question types coupled with variable scene complexity demand flexible adaptation, and (3) the model compensates for missing contextual priors through increased cross-modal confirmation. These contrasting statistical profiles establish the empirical foundation for understanding how grounding availability shapes information decomposition strategies.

Table 7: Dataset-specific DAM weight statistics across all question types. Mean $\pm$ standard deviation reported for each atom. All values are aggregated from multi-seed experiments (seeds: 42, 1024, 2021, 3407).

| Dataset | $\beta_{U_1}$ (Point) | $\beta_{U_2}$ (Image) | $\beta_R$ (Redundancy) | $\beta_S$ (Synergy) |
|---|---|---|---|---|
| SQA3D | $0.237 \pm 0.061$ | $0.252 \pm 0.087$ | $0.245 \pm 0.041$ | $0.266 \pm 0.090$ |
| ScanQA | $0.234 \pm 0.231$ | $0.207 \pm 0.138$ | $0.295 \pm 0.142$ | $0.263 \pm 0.228$ |
| **Overall** | $0.235 \pm 0.181$ | $0.226 \pm 0.122$ | $0.274 \pm 0.115$ | $0.265 \pm 0.184$ |

**Question-Type Adaptation: Cognitive Alignment vs. Compensatory Behavior.** Tab. 8 decomposes weight statistics by question type, revealing how reasoning requirements interact with grounding availability. For SQA3D, we observe systematic patterns that align with cognitive reasoning principles: *What* questions emphasize modality-specific uniqueness ($\beta_{U_1} = 0.273 \pm 0.079$, $\beta_{U_2} = 0.279 \pm 0.078$) for precise object identification; *How* questions amplify synergy ($\beta_S = 0.360 \pm 0.085$) as spatial relationships fundamentally require emergent information from joint 3D-2D processing; *Can* questions prioritize image uniqueness ($\beta_{U_2} = 0.344 \pm 0.057$) and redundancy

($\beta_R = 0.320 \pm 0.016$) for visual affordance assessment and cross-modal confirmation. These patterns directly reflect the inherent information requirements of each question type when operating under proper grounding conditions.

ScanQA exhibits markedly different patterns that illuminate compensatory adaptation mechanisms. Most strikingly, *How* questions show inverted emphasis: high point cloud uniqueness ($\beta_{U_1} = 0.369 \pm 0.249$) and suppressed synergy ($\beta_S = 0.122 \pm 0.124$), contrasting sharply with SQA3D's synergy-dominant pattern ($\beta_S = 0.360 \pm 0.085$). This inversion reveals that without situational anchoring, spatial reasoning cannot rely on emergent cross-modal interactions but must instead substitute explicit geometric feature extraction. Similarly, *What* questions demonstrate elevated redundancy ($\beta_R = 0.290 \pm 0.141$) compared to SQA3D ($\beta_R = 0.201 \pm 0.044$), indicating increased cross-modal confirmation when contextual priors are unavailable. These compensatory patterns are not artifacts but rather evidence of the model's adaptive capacity to reconfigure information integration strategies under impoverished input conditions.

Table 8: Question-type-specific DAM weight statistics. Mean $\pm$ standard deviation reported for each atom within each question type. N denotes the number of samples. "–" indicates question types not present in the dataset.

| Dataset | Question Type | $\beta_{U_1}$ (Point) | $\beta_{U_2}$ (Image) | $\beta_R$ (Redundancy) | $\beta_S$ (Synergy) | N |
|---|---|---|---|---|---|---|
| SQA3D | What | $0.273 \pm 0.079$ | $0.279 \pm 0.078$ | $0.201 \pm 0.044$ | $0.247 \pm 0.085$ | 4,232 |
| | Is | $0.216 \pm 0.032$ | $0.290 \pm 0.074$ | $0.260 \pm 0.023$ | $0.234 \pm 0.063$ | 2,448 |
| | How | $0.231 \pm 0.043$ | $0.188 \pm 0.068$ | $0.221 \pm 0.036$ | $0.360 \pm 0.085$ | 2,048 |
| | Can | $0.177 \pm 0.033$ | $0.344 \pm 0.057$ | $0.320 \pm 0.016$ | $0.159 \pm 0.034$ | 1,124 |
| | Which | $0.213 \pm 0.033$ | $0.293 \pm 0.057$ | $0.281 \pm 0.025$ | $0.214 \pm 0.043$ | 960 |
| | Other | $0.222 \pm 0.043$ | $0.285 \pm 0.070$ | $0.267 \pm 0.028$ | $0.226 \pm 0.059$ | 2,232 |
| ScanQA | What | $0.223 \pm 0.229$ | $0.212 \pm 0.145$ | $0.290 \pm 0.141$ | $0.275 \pm 0.242$ | 11,488 |
| | Is | $0.302 \pm 0.214$ | $0.314 \pm 0.109$ | $0.259 \pm 0.120$ | $0.125 \pm 0.140$ | 16 |
| | How | $0.369 \pm 0.249$ | $0.310 \pm 0.115$ | $0.200 \pm 0.120$ | $0.122 \pm 0.124$ | 1,104 |
| | Can | | | – | | 0 |
| | Which | $0.263 \pm 0.264$ | $0.212 \pm 0.083$ | $0.292 \pm 0.141$ | $0.232 \pm 0.169$ | 92 |
| | Other | $0.230 \pm 0.223$ | $0.180 \pm 0.116$ | $0.323 \pm 0.139$ | $0.268 \pm 0.207$ | 6,000 |

**Reconciling Dataset-Specific Patterns:** The contrasting behaviors between SQA3D and ScanQA do not undermine but rather strengthen our interpretability framework through three complementary insights. *First*, SQA3D patterns establish the cognitive grounding for our general conclusions by demonstrating how information atoms manifest when the model operates under ideal conditions: with sufficient situational context to directly address question semantics through proper atom utilization. The tight distributions and cognitively-aligned adaptations (synergy for spatial reasoning, uniqueness for identification) validate that DAM learns decomposition strategies reflecting inherent information requirements. *Second*, ScanQA patterns reveal the framework's diagnostic capacity by exposing compensatory mechanisms when grounding is absent. The inverted patterns and wider distributions are not failures but transparent signatures of adaptive reconfiguration, the model substituting geometric analysis for cross-modal synergy when situational anchoring is unavailable. This exposure of compensatory strategies is precisely what interpretable systems should provide. *Third*, the dataset-specific differences validate a fundamental claim: by decomposing information atoms explicitly, ATOM reveals not just static reasoning patterns but dynamic adaptation to varying information availability. Our interpretation aligns more closely with SQA3D because it represents unconfounded reasoning. ScanQA patterns, while demonstrating valuable robustness, are inherently confounded by simultaneous needs to resolve missing context and answer questions. These complementary roles, SQA3D establishing cognitive principles, ScanQA demonstrating adaptive robustness, provide comprehensive evidence that DAM weight distributions offer genuine interpretability across operational conditions.

## F  USER STUDY

**Participants**    Participants were all compensated at the local minimum wage rate. We recruited five participants, aged from 22 to 25 ($M = 24$, $SD = 1.22$), via university emails and posts on social media. All participants were pursuing or held PhD degrees or above in Computer Science, possessed normal vision with no color deficiency, and had at least three years of research experience in AI/ML. To assess domain expertise, we employed a 5-point Likert scale (1 = no experience; 5 = very professional with multiple top-tier publications). Participants reported moderate-to-high familiarity with computer vision ($M = 3.20$, $SD = 1.10$), 3D vision ($M = 3.40$, $SD = 0.89$), and interpretable AI ($M = 3.40$, $SD = 1.67$). When asked to rate the importance of model interpretability for developing or deploying AI solutions in complex 3D environments, participants demonstrated strong consensus on its value ($M = 4.20$, $SD = 0.84$), with four out of five rating it as important or very important.

## G  MORE QUALITATIVE RESULTS

Additional qualitative results illustrating our model's grounding and reasoning capabilities are shown in Fig. 7 and Fig. 8. These examples highlight ATOM's versatility across diverse question types, including locating objects, identifying their attributes or states, counting instances within a scene, and addressing yes/no queries that require commonsense reasoning. The results further demonstrate that ATOM remains robust in complex environments, even when the description and question conflict (e.g., referring to different objects or providing misleading context) or when the description is overly intricate.

In addition, we provide corresponding DAM weight distributions for selected samples from the SQA3D dataset (see Fig. 9). These sample-level analyses reveal a critical insight: conventional fusion methods that rely primarily on redundant cross-modal information fundamentally misunderstand multimodal interactions in 3D QA. Our DAM weights demonstrate that modality-specific uniqueness and cross-modal synergy often contribute substantially to correct predictions—information that black-box approaches systematically overlook. For instance, geometric uniqueness and visual uniqueness become crucial for *What* questions requiring precise spatial localization and object identification. Synergy dominates *How* questions that demand emergent spatial reasoning about object relationships, distances, and quantities. Redundancy proves essential for questions requiring consistent cross-modal confirmation, such as object functionality recognition or directional navigation queries where both modalities must agree on spatial relationships. This validates ATOM's information-theoretic foundation: by explicitly modeling and adaptively weighting these distinct information atoms, our framework captures the full spectrum of multimodal interactions necessary for robust 3D spatial reasoning.

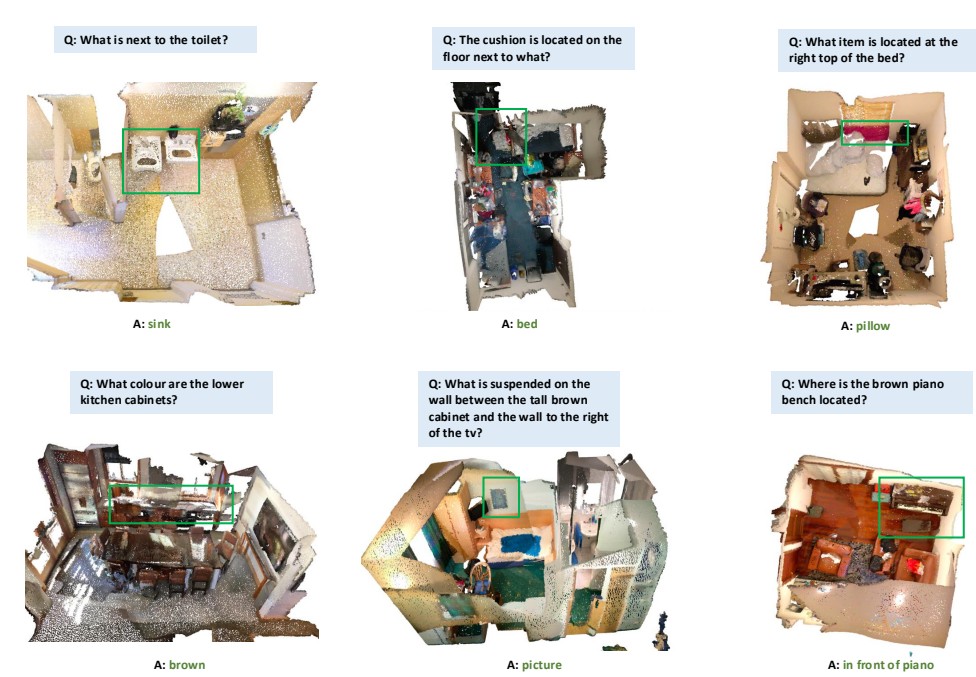

Figure 7: More qualitative examples from the ScanQA dataset. Green rectangles highlight the objects identified by ATOM as critical for answering the question correctly.

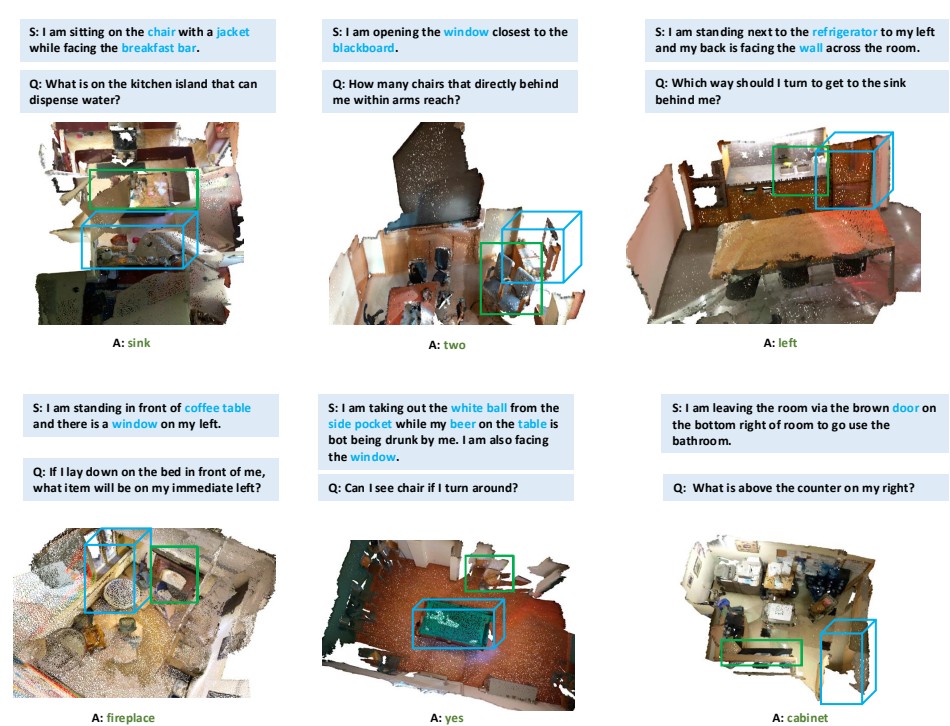

Figure 8: More qualitative examples from the SQA3D dataset. Blue bounding cubes denote the agent's situated position, while green boxes highlight the objects identified by ATOM as critical for answering the question correctly.

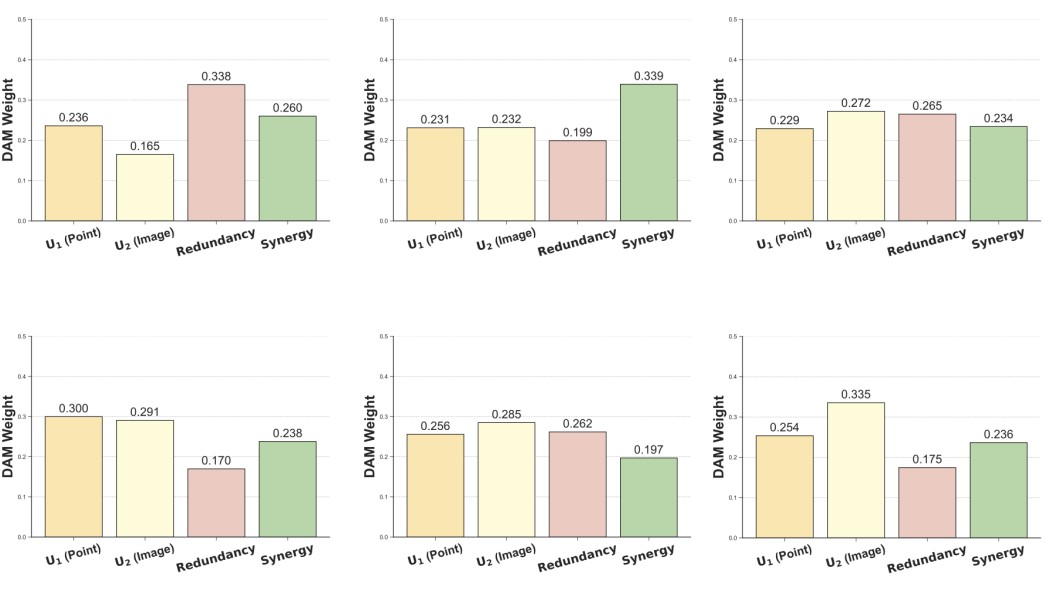

Figure 9: Corresponding DAM weight distribution for the selected example from the SQA3D dataset.