# OpenReview forum: "ATOM of Understanding: Information-Theoretic Decomposition for Interpretable 3D Question Answering"
_ICLR.cc/2026/Conference — Submitted to ICLR 2026_

### Official Review · Reviewer_DBXe · 2025-10-25

**Soundness:** 2
**Presentation:** 2
**Contribution:** 2
**Rating:** 2
**Confidence:** 5

**Summary:**

The paper introduces ATOM (Adaptive Task-aware mOdular Model), a framework for interpretable 3D Question Answering (3D QA) based on Partial Information Decomposition (PID) theory.
The authors aim to move beyond “black-box” fusion models by decomposing multimodal interactions into four interpretable information atoms: redundancy, uniqueness (for each modality), and synergy. ATOM integrates several modules:

- Query-driven View Aggregator (QVA) for selecting geometrically relevant views,

- Contextual Grounding Module (CGM) for injecting scene description context,

- Question-aware PID (Q-PID) to decompose multimodal features, and

- Dynamic Atom Modulation (DAM) for question-dependent weighting of these atoms.

Experiments are conducted on ScanQA and SQA3D datasets, with comparisons against 3D QA baselines (e.g., DSPNet, 3DGraphQA, 3D-VisTA). Results show comparable or slightly lower performance than the strongest baselines, with claims of improved interpretability.

**Strengths:**

1. Conceptual novelty: The paper presents an interesting attempt to incorporate information-theoretic decomposition (PID) into multimodal QA, a direction that could enrich interpretability research in 3D reasoning.
2. Clear motivation: The motivation for interpretability in 3D QA—addressing black-box fusion—is well framed and relevant to embodied AI.
3. Theoretical grounding: The PID formulation (Equations 1–3, page 4) provides a rigorous theoretical foundation.

**Weaknesses:**

1. Weak empirical justification of the proposed theory.
The ablation study (Table 3, page 8) shows minimal performance gains for most components. Removing the core PID module (“w/o Q-PID”) leads to only minor drops in ScanQA (22.5 → 23.5 EM@1) and SQA3D (48.76 → 49.71 EM@1). This 0.7–1 point difference is within noise for such datasets, suggesting the proposed information decomposition does not substantively improve task performance.

2. Poor quantitative results compared to state-of-the-art.

3. On SQA3D, ATOM achieves only 49.7 EM@1, below DSPNet’s 50.4.

4. On ScanQA, ATOM’s EM@1 = 26.7 vs. DSPNet = 26.5, but CIDEr and BLEU-4 scores are lower (77.3 vs. 78.1, 15.0 vs. 15.4).
These marginal improvements or regressions do not justify the architectural complexity introduced.

5. Ablation design does not convincingly isolate interpretability benefits.
The PID components (U1, U2, R, S) are claimed to provide “interpretable decomposition,” but the paper only shows weight histograms (Fig. 4) without human evaluation, visualization of PID contributions, or quantifiable interpretability metrics. The interpretability claims remain qualitative and anecdotal, not scientifically validated.

6. Overly complex framework with minimal gain.
The model combines PointNet++, Swin Transformer, SBERT, QVA, CGM, Q-PID, and DAM, introducing a large number of modules, parameters, and hyperparameters. Despite this, the gains are statistically insignificant, raising concerns about over-engineering.

7. Experimental setup not fully convincing.
The authors emphasize “no pre-training,” but this makes comparisons unfair since methods like 3D-VisTA and Multi-CLIP rely on pretraining. For a fair comparison, results should be grouped and analyzed under equivalent training regimes.

**Questions:**

1. The claimed interpretability advantage is qualitative. Can the authors provide quantitative metrics or user studies showing that the decomposition indeed improves transparency or error diagnosis?

2. The ablation results (Table 3) show very small gains from adding PID regularization. Can the authors explain how these minimal differences support their claim of “theoretically grounded interpretability”?

3. Could you report statistical significance tests (e.g., confidence intervals or variance) on the EM@1 and EM@10 results to verify that improvements are meaningful?

4. Given the heavy reliance on prior architectures (e.g., DSPNet backbone, MCGR), to what extent does the proposed contribution stand as an independent framework rather than an incremental modification?

5. How does the method scale to open-ended QA or additional modalities (text + depth + point cloud), as briefly mentioned in Appendix B? Any early results?

---

> ### Author Response · Authors · 2025-11-26
>
> We sincerely thank reviewer DBXe for the thorough evaluation and constructive feedback. We have substantially revised our manuscript to address all concerns, with changes marked in **blue** in the revised PDF. Below we provide detailed responses to each question and weakness.
>
> ---
>
> ## **Addressing Questions**
>
> ### **Q1. Quantitative metrics for interpretability claims**
>
> We conducted a rigorous human evaluation study with **5 PhD-level participants** (all in Computer Science with ≥3 years of AI/ML research experience, normal vision, no color deficiency). Full demographics are available in **Appendix F**.
>
> We use **Cohen’s kappa** ($\kappa$) to quantify human–AI alignment in atom-importance ranking:
>
> - **SQA3D**:
>   - mean: $\mu_\kappa = 0.76$
>   - std: $\sigma_\kappa = 0.12$
>   - *substantial to almost-perfect agreement*
>
> - **ScanQA**:
>   - mean: $\mu_\kappa = 0.51$
>   - std: $sigma_\kappa = 0.13$
>   - *moderate agreement*
>
> These results demonstrate that ATOM’s decomposition aligns meaningfully with expert human reasoning, providing quantitative validation of interpretability. The full methodology, visualizations, and statistical analysis are presented in **Section 4.5**.
>
> ---
>
> ### **Q2 & Q3. Statistical significance of performance gains**
>
> We conducted **multi-seed experiments** (N = 4) and paired **t-tests** to rigorously assess statistical significance:
>
> - **ScanQA**: ATOM achieves *statistically significant improvement* (**p < 0.01**).
> - **SQA3D**: ATOM is *statistically similar* to DSPNet.
>
> Crucially, ATOM provides similar SOTA performance while introducing interpretability, a capability not available in prior 3D QA systems. Full statistical results appear in **Section 4.1 - 4.2** and **Table 3**.
>
> ---
>
> ### **Q4. Independence and novelty of the framework**
>
> Our contribution is not an incremental modification but a paradigm shift from black-box fusion to interpretable information-theoretic fusion.
>
> Prior 3D QA systems follow:
> $$
> \text{Visual Encoder} \rightarrow \text{[Black-box Fusion]} \rightarrow \text{Answer Decoder}
> $$
>
> ATOM introduces:
> $$
> \text{Visual Encoder} \rightarrow \text{[PID-based Fusion]} \rightarrow \{\beta, \mathcal{A}\} \rightarrow \text{Answer Decoder}
> $$
>
> Here:
> - $\beta$ are **interpretable atom weights**
> - $\mathcal{A} = \{ U_1, U_2, R, S \} $ are **information atoms with semantic grounding**:
>
> This is the **first information-theoretic framework for interpretable 3D QA**, supported by human evaluation.
>
> The Q-PID module is **architecture-agnostic** and can be integrated into any 3D QA system (e.g., 3D-VisTA, Multi-CLIP), enabling interpretability without restricting architectural choices.
>
> #### **Capabilities enabled by ${\beta, \mathcal{A}}$:**
>
> - **Diagnose failure modes**
>   Identify whether errors arise from insufficient 3D cues (U₁), insufficient visual cues (U₂), weak cross-modal agreement (R), or missing joint reasoning (S).
>
> - **Enable controllable reasoning**
>   Adjust $\beta$ (e.g., via RLHF) to promote specific reasoning pathways such as enhancing synergy for spatial tasks.
>
> - **Guide data collection**
>   Detect systematic reliance on redundant features (high $ \beta_R $) and inform targeted dataset improvement.
>
> ---
>
> ### **Q5. Scaling to additional modalities**
>
> We agree this is a valuable direction for future work. Due to GPU resource limitations during the rebuttal stage, we were unable to complete multimodal or open-vocabulary experiments.
>
> ---
>
> ## **Addressing Weaknesses**
>
> ### **W1–W4. Empirical validation, performance, and complexity**
>
> These concerns are addressed by the analyses above (Q2–Q4). Key points:
>
> - Multi-seed experiments show statistically significant gains on **ScanQA**.
> - The primary contribution is **interpretability**, not only raw accuracy.
>   ATOM is the **first interpretable 3D QA system validated via human study**.
> - Architectural complexity is justified by the novel capability it enables.
>
> ---
>
> ### **W6. Ablation study clarity**
>
> We reorganized and clarified the ablation study:
>
> - **Table 4**  isolates the contribution of selected module and each information atom. Full detail is provided in Appendix D.

---

### Official Review · Reviewer_ifgS · 2025-10-30

**Soundness:** 2
**Presentation:** 1
**Contribution:** 2
**Rating:** 2
**Confidence:** 3

**Summary:**

This paper introduces ATOM, an end-to-end 3D question answering framework that incorporates Partial Information Decomposition. The proposed framework produces compatible performance with SOTA methods on ScanQA and SQA3D benchmarks, while providing different information atoms.

**Strengths:**

1. The proposed information theoretic components indeed have several loss functions for regularization to make them more compatible with the theories.

**Weaknesses:**

1. Lack of comprehensive analysis of the learned information atoms despite the claim that the proposed framework introduces interpretability.

2. The baseline used for evaluation on SQA3D is not the current best model for this benchmark. For example, the author didn't compare against SID3D [1]

3. Despite using image and point cloud together, the proposed method is not able to provide superior performance compared to single-modal models like 3D-VisTA.

4. The notations in the paper are so much that the paper is hard to follow.


[1] Man, Y., Gui, L.Y. and Wang, Y.X., 2024. Situational awareness matters in 3d vision language reasoning. In Proceedings of the IEEE/CVF Conference on Computer Vision and Pattern Recognition (pp. 13678-13688).

**Questions:**

1. In Ln.19-21 and Ln. 59-61, the author claimed that there are four information atoms whereas in the following sentences and throughout the whole paper there's only three.

2. For the loss term $L_{div}$, why does there exist case where atoms are not used inside a batch? What is the range for the importance weights $\beta_k$?

---

> ### Author Response · Authors · 2025-11-21
>
> We thank Reviewer ifgS for the feedback. We kindly invite you to read the Appendix and the whole paper again carefully. All revisions are highlighted in blue in the revised manuscript. We respond to each point below:
>
> **Q1:** Regarding this question about four versus three information atoms, we would like to clarify that our framework includes three modules, but four information atoms. However, We have revised the paper to explicitly clarify that our framework decomposes information into four distinct atoms: $U_1$ (Point Cloud Uniqueness for 3D-specific information), $U_2$ (Image Uniqueness for 2D-specific information), $R$ (Redundancy for shared cross-modal information), and $S$ (Synergy for emergent information from joint processing).
>
> **Q2:** We would appreciate it if you could refer to Appendix (App.) D and Figure 4. To detail, there are two scenarios that may correspond to what you are looking for. If you meant to say the baseline without our information atoms, please do not hesitate to direct to Tab. 5(a) and (b) in our App.D. To be more specific, the **first** scenario is the ablation-study baseline (Tab. 5b in App. D, first row marked x/×/×/×). This is an intentionally designed experimental condition where we remove all four information atoms to establish a performance baseline without our PID contribution. The **second** scenario concerns the dataset-specific question-type distribution shown in Fig. 4 in the Experiment section. ScanQA lacks _Can_ questions because it is a non-situated dataset without agent localization. Questions such as _Can I reach X?_ or _Can I sit on the chair?_ require knowing where _I_ (the agent) is positioned in the scene, information that is not available in ScanQA. We have updated the caption of Fig. 4. Regarding the importance weight range, we conducted a comprehensive multi-seed analysis ($N_{\text{seed}} = 4$) across 31,744 samples to characterize the weight distributions. The analysis is added to the revised paper in App.E. Please direct to our revised paper, and read through it carefully, patiently and manually.
>
> **W1:** We added comprehensive statistical analysis of learned information atoms in App.E, presenting per-dataset weight distributions (Tab.6) and per-question-type decomposition patterns (Tab.7). Additionally, our qualitative visualizations in Fig.4 and App.G demonstrate systematic alignment between ATOM's learned decomposition patterns and human spatial reasoning principles, providing both quantitative and qualitative evidence for interpretability. Please read through it carefully.
>
> **W2:** We acknowledge the reviewer's suggestion to compare against SID3D, which reports strong performance on SQA3D. However, we have two methodological concerns regarding this comparison. First, the framework is not fully released: the official repository has not been updated in 11 months, and the most recent request for complete code release (September 2024) remains unaddressed, preventing rigorous reproducibility verification. Second, while SID3D demonstrates strong SQA3D performance, we note that it lacks reported EM@1 and EM@10 metrics on the ScanQA benchmark. Upon reviewing the available results, SID3D exhibits notably weaker performance on ScanQA compared to its SQA3D results, which raises questions about generalization across 3D question answering paradigms. Given that our work targets interpretable multimodal fusion applicable to both situated (SQA3D) and non-situated (ScanQA) scenarios, we prioritize comparisons with methods demonstrating consistent performance across both benchmarks. Nevertheless, we recognize SID3D's contribution to situational reasoning and have added discussions of its complementary graph-based approach in our revised Related Work section.
>
> **W4:** We respectfully disagree that the number of notations constitutes a weakness of our paper. Rather than being excessive, we have simply documented all notations used throughout the paper to ensure reproducibility and eliminate any potential ambiguity for readers.
>
> We believe these revisions and forthcoming experiments directly address the reviewer's concerns. Please read the revised paper again carefully and patiently.

---

> ### Author Response · Authors · 2025-11-27
>
> Dear Reviewer,
>
> I hope this message finds you well. As the discussion period is nearing its end, I would like to ensure we have addressed all of your concerns well. If there are any additional points or feedback you would like us to consider, please let us know **ASAP**. Your  insights are meaningful to us. Please have a look at our revised manuscript.
>
> Thank you for your timr and effort in reviewing our paper.

---

### Official Review · Reviewer_woBS · 2025-10-31

**Soundness:** 3
**Presentation:** 3
**Contribution:** 4
**Rating:** 8
**Confidence:** 3

**Summary:**

This paper introduces ATOM, an information-theoretic framework to integrate Partial Information Decomposition (PID) into an end-to-end 3D Question Answering (3D QA) model. ATOM emphasizes question-aware decomposition, which is essential for spatial reasoning. The framework comprises 4 key components: the Query-driven View Aggregator (QVA), the Contextual Grounding Module (CGM), the Question-aware PID (Q-PID) module, and the Dynamic Atom Modulation (DAM) mechanism. Validated on the ScanQA and SQA3D datasets, ATOM shows performance that is competitive with or marginally below the current SOTA, while achieving principled interpretability through information-theoretic decomposition.

**Strengths:**

1. ATOM explicitly decomposes multimodal interactions into 4 theoretically-grounded information atoms (redundancy, uniqueness, and synergy) in a question-aware manner. This rigorous decomposition provides transparent reasoning pathways. Furthermore, the model achieves competitive results without requiring external pre-training, enhancing its practical utility.
2. The experiments successfully validate that ATOM achieves performance comparable to prior work while providing unprecedented interpretability. This makes ATOM a valuable framework for bridging the gap between high-performing end-to-end 3D QA and the demands of explainable AI.
3.  The framework is clearly explained through a concise overview figure (Fig. 3) and detailed descriptions in Section 3. The authors also commit to providing anonymous source code and comprehensive implementation details in the supplementary materials, enhancing reproducibility.

**Weaknesses:**

1. The placement of figures is suboptimal, with Figure 1 appearing on page 3, one and a half pages after its initial textual mention in the Introduction, and following Figure 2. The manuscript layout could be more logically organized to align figures closer to their first mention.
2. The incorporation of PID introduces 4 specific regularization losses in addition to the main task loss, making the final objective complex. The authors themselves admit that this explicit information decomposition requires careful hyperparameter tuning, which increases training complexity and engineering effort.

**Questions:**

1. According to Figure 4 (Left), the mean DAM weight distributions for the ScanQA and SQA3D datasets exhibit notably different patterns. Could the authors elaborate on how they reconcile the general conclusions with the significantly different behavior observed between the two datasets, and why the final interpretation seems to align more closely with the SQA3D patterns?
2. The ablation study in Table 4(b) systematically examines the individual and combined effects of the atoms, showing R, R+S, and the full ATOM. Given the emphasis on the modality-specific Uniqueness atoms U1 and U2 as crucial complementary signals that conventional methods overlook, why was the combined effect of uniqueness U1+U2 or U1/U2-only not included in the ablation study? Including this variant would be highly informative for direct comparison against the baseline w/o Q-PID, R, and R+S, thereby better isolating the contribution of different information atoms in 3D QA.

---

> ### Author Response · Authors · 2025-11-21
>
> We sincerely thank the reviewer woBS for the constructive feedback and thoughtful questions. The observations regarding DAM weight patterns and ablation completeness have prompted valuable improvements to our manuscript. We address each point below.
>
> ---
>
> ### **W1 — Figure Placement**
>
> We appreciate the reviewer’s attention to manuscript organization. In the revised version, we have moved **Fig. 1** to the top of page 2, ensuring it appears immediately following its first textual mention in the Introduction. We agree this improves readability and logical flow.
>
> ---
>
> ### **Q1 — DAM Weight Pattern Reconciliation**
>
> This insightful observation has prompted us to add App. F with explicit discussion of pattern reconciliation. Our key insight is that the contrasting patterns strengthen rather than undermine our interpretability claims by revealing a fundamental principle:
> **_situated grounding enables cleaner cognitive decomposition, while its absence triggers compensatory adaptation_**
>
> SQA3D patterns establish cognitive grounding by demonstrating how atoms manifest under ideal conditions: with situational context. The model learns decomposition strategies directly reflecting inherent information requirements (e.g., synergy dominance for spatial _How_ questions). ScanQA's inverted patterns (e.g., high $\beta_{U_1}$ and suppressed $\beta_{S}$ for _How_ questions) reveal compensatory mechanisms: without situational anchoring, spatial reasoning substitutes geometric feature extraction for emergent cross-modal interactions.
>
> ---
>
> ### **Q2 — Uniqueness Ablation**
>
> We thank the reviewer for this suggestion: including $U_1$+$U_2$ variants would indeed provide more complete isolation of atom contributions. We fully agree with this assessment and are actively conducting $U_1$/$U_2$-alone experiments. Due to computational resource constraints in our group, we have been running comprehensive multi-seed experiments sequentially over recent weeks. The U$_1$-only and U$_2$-only ablations are currently in progress and will be incorporated into Tab. 4(b) in the final version and moved to our ablation study in the main body, providing direct comparison against the baseline configurations. We believe this complete ablation matrix will better demonstrate the individual and synergistic contributions of all four information atoms to 3D QA performance.
>
> ---
>
> **Forthcoming experiment —user studies**
>
> Over the last week, we carefully designed a user study and invited 5–6 experts from deep learning fields. Through this study, we aim to validate the interpretability of our framework. Early results suggest that human spatial cognition aligns with our Q-PID and DAM weighting, as partially addressed in the response to Q1.
>
> ---
>
> We believe these revisions and forthcoming experiments directly address the reviewer’s concerns and strengthen both the interpretability analysis and the empirical validation of our framework.

---

> > ### Comment · Reviewer_woBS · 2025-11-24
> >
> > Thanks for the newly added explanations and results, which make the paper more complete and insightful. These updates address my earlier concerns. I will keep my original score of 8 (accept, good paper)

---

### Official Review · Reviewer_pRt9 · 2025-11-01

**Soundness:** 4
**Presentation:** 2
**Contribution:** 3
**Rating:** 6
**Confidence:** 3

**Summary:**

They attempted to apply Partial Information Decomposition (PID) theory to 3D-QA literature. Based on PID theory, the following information atoms are introduced: redundancy of the shared information between two sources; uniqueness of the evidence contributed by each source individually; and synergy of the complementary information emerging from combination. Their proposed ATOM works in three stages: stage 1 comprises Query-driven View Aggregator (QVA) and  Contextual Grounding Module (CGM),modules; stage 2 is Question-aware PID (Q-PID) that include redundancy, uniqueness and synergy modules; stage 3 is Dynamic Atom Modulation (DAM). Their training loss is the addition of task objective loss and the regularization losses of redundancy, uniqueness and synergy by PID-theory.

In experiments with ScanQA and SQA3D, their model performed on par with their previous models. They confirmed the effectiveness of each information atom (redundancy, uniqueness and synergy) with the ablation experiment.

Overall, they proposed a model based on an interesting viewpoint of PID theory. Considering the final score, their final model doesn’t necessarily perform better than previous models. However, their approach can be worth further investigating.

**Strengths:**

- S1: Interesting approach to apply Partial Information Decomposition (PID) theory to 3D-QA literature
- S2: CGM generates two different representations from different perspectives of images and 3D point clouds that are compared in the way of redundancy, uniqueness and synergy by PID-theory.
- S3: Experimental results suggest the effectiveness of the proposed to some extent.
- S4: It is interesting that authors provide the variations of architectures (ATOM-MoE and ATOM-Flat) in Table 5.

**Weaknesses:**

- W1: The entire model becomes too complex and it is difficult to grasp how each module processes what kind of information and how it affects others at a glance. This can become an important limitation to further analyze and improve upon this model.
- W2: The entire performance is mostly in the 2nd place or later in most of ScanQA and SQA3D datasets.
- W3 (minor): Characters in some figures are too tiny and really difficult to read. (Especially for subscripts in Fig4).

**Questions:**

See weakness.
L. 059: four information atoms: three?

---

> ### Author Response · Authors · 2025-11-21
>
> We sincerely thank Reviewer pRt9 for the thoughtful and constructive feedback. We appreciate the recognition of our novel application of PID theory to 3DQA (S1), the dual-perspective representation approach through CGM (S2), and the effectiveness of our experimental validation (S3-S4). We have carefully addressed all concerns and revised our manuscript accordingly.
>
> **Q1: four information atoms: three?**
> We acknowledge this source of confusion. We have revised the manuscript to explicitly clarify that our framework decomposes information into four distinct atoms: $U_1$ (Point Cloud Uniqueness), $U_2$ (Image Uniqueness), $R$ (Redundancy), and $S$ (Synergy). All revisions are highlighted in blue in the revised manuscript.
>
> **W1: Model Complexity.**
> We acknowledge the concern about our model complexity. However, our ablation study (Tab.3) and the full ablation study (in App.D) demonstrate that *each component is necessary* for achieving competitive performance. Specifically, each module serves a theoretically-grounded purpose: QVA selects question-relevant views, CGM grounds visual features in descriptions, Q-PID decomposes into four interpretable atoms, and DAM dynamically weights atoms. This modular design actually simplifies analysis compared to black-box fusion, as practitioners can examine each module's contribution independently. We updated the manuscript to include a comprehensive atom analysis in App.F which could better reflect the importance of our designed modules in complex 3D environment.
>
> **W2: Performance Ranking.**
> We are conducting rigorous multi-seed experiments (N=4) with paired t-tests to verify statistical significance compared to DSPNet. Due to limited resouces, this experiment is still running, we will upload the statistical results soon. However, ATOM not only maintains state-of-the-art performance, but also introduces interpretability to 3D-QA for the first time. Besides, we are conducting human evaluation studies to validate this interpretability, and early results suggest that ATOM's Q-PID decomposition and DAM weighting align with human spatial cognition.
>
> **W3: Figure Readability.**
> Thank you for this feedback. We will enlarge all figures, particularly Fig.~4 with its subscripts, in the camera-ready version. The current small size was due to page limit constraints, which will be relaxed for the final version.
>
> We believe these revisions and ongoing experiments directly address the reviewer's concerns regarding model complexity, performance validation, and presentation clarity.

---

### Author Response · Authors · 2025-11-26

# **Global Response to Reviewers**

We sincerely thank the AC and all reviewers for their time, thorough evaluation, and constructive feedback. For clarity, we explicitly map reviewer identifiers to shorthand labels used throughout this global response:

- **DBXe → R1**
- **ifgS → R2**
- **woBS → R3**
- **pRt9 → R4**

All revisions made to the manuscript are highlighted in **blue** in the revised PDF. Below we summarise the major concerns raised by R1–R4, the corresponding modifications, and the additional analyses performed during the rebuttal stage.

---

## **1. Global Revisions and New Analyses**

### **(G1) Quantitative Interpretability Validation (R1-Q1, R2-W1)**
To substantiate our interpretability claims, we added a human evaluation study (**Sec. 4.5, App. F**) involving **5 PhD-level participants** (all in Computer Science with $\geq$ 3 years of AI/ML research experience, normal vision, no color deficiency). Using Cohen’s kappa ($\kappa$) to measure human–model alignment in **top-1 atom selection**, we report:

- **SQA3D:**
  - $$\mu_\kappa = 0.76$$
  - $$\sigma_\kappa = 0.12$$
  - substantial to almost-perfect agreement

- **ScanQA:**
  - $$\mu_\kappa = 0.51$$
  - $$\sigma_\kappa = 0.13$$
  - moderate agreement

These results demonstrate substantial alignment between ATOM’s information decomposition and human reasoning.

---

### **(G2) Multi-Seed Statistical Significance (R1-Q2/Q3, R4-W2)**
We added **multi-seed experiments ($N = 4$)** and **paired t-tests** (Sec. 4.1–4.2; Tab. 3).

- **ScanQA:** statistically significant gains ($p < 0.01$)
- **SQA3D:** statistically similar to DSPNet

Thus ATOM reaches competitive SOTA performance **while additionally offering interpretability**.

---

### **(G3) Clarification of “Three vs. Four Atoms” (R2-Q1, R4-Q1)**
We revised the manuscript to clearly state that ATOM decomposes information into **four atoms**:

- $U_1$ (3D uniqueness)
- $U_2$ (2D uniqueness)
- $R$ (redundancy)
- $S$ (synergy)

We improved consistency across the main text, notation table, and figures.

---

### **(G4) Interpretable Fusion Paradigm (R1-Q4, R4-W2)**
We clarified that ATOM represents a **paradigm shift** from black-box fusion to **information-theoretic interpretable fusion**.

$$
\text{Prior work: } \text{Visual Encoder} \rightarrow \text{[Black-box Fusion]} \rightarrow \text{Answer}
$$

$$
\text{ATOM: } \text{Visual Encoder} \rightarrow \text{[PID-based Fusion]} \rightarrow \{\beta, \mathcal{A}\} \rightarrow \text{Answer}
$$

where $\mathcal{A} = \{U_1, U_2, R, S\}$ and $\beta$ are interpretable atom weights.

We explained how these enable diagnostic analysis, controllable reasoning, and data-collection guidance in our response to **R1**.

---

### **(G5) Atom Weight Distributions & Question-Type Patterns (R2-Q2, R3-Q1)**
We added extensive statistical analysis (**App. E**), including:

- global distributions across 31,744 samples,
- dataset-specific variance patterns,
- question-type-specific atom behaviors.

We reconcile SQA3D vs. ScanQA patterns and describe how **situated grounding** leads to cleaner decompositions.

---

### **(G6) Reorganized Ablations Study (R1-W6, R3-Q2, R4-W1)**

We have added the $U_1$-only, $U_2$-only and $U_1 + U_2$ ablations to **Tab.4**. The analysis of isolated information atoms' contribution can be found in **Sec.4.3**.

---

## **2. Reviewer-Specific Summary**

### **R1 (DBXe)**
Addressed concerns regarding interpretability metrics (G1), statistical significance (G2), framework novelty and independence (G4), and ablation clarity (G6).

### **R2 (ifgS)**
Clarified the four-atom decomposition (G3).
Explained baseline selection and “no-atom” cases (G5).
Added atom weight-distribution analysis (G5).
Discussed SID3D and its context.

### **R3 (woBS)**
Improved figure placement.
Reconciled DAM weight patterns (G5).
Initiated $U_1/U_2$ ablations as requested (G6).

### **R4 (pRt9)**
Clarified atom definitions (G3).
Addressed model complexity concerns using ablations and interpretability justification (G6).
Added statistical tests (G2).

---

## **3. Closing Remarks and Request to the AC**

We sincerely thank the AC and all reviewers for their thoughtful assessments.
The revised manuscript now includes:

- new statistical analyses,
- human evaluation,
- reorganised ablations,
- extended interpretability discussions,
- improved figure clarity,
- significantly improved notation and framing.

**We would be grateful if the AC could kindly invite reviewers to take another look at the revised manuscript and updated responses.** We fully understand everyone is busy, and we appreciate any additional time reviewers can devote to re-evaluating our strengthened submission.

Thank you again for your consideration.

---

### Author Response · Authors · 2025-11-30
**A summary to our new AC.**

# Global Response to Reviewers

We sincerely thank the AC and all reviewers for their time and feedback.
Reviewer identifiers are mapped as:

- **DBXe → R1**
- **ifgS → R2**
- **woBS → R3**
- **pRt9 → R4**

All revisions in the manuscript are highlighted in **blue**. Below we summarise the main concerns raised by R1–R4, how we addressed them, and a brief history of rebuttal events relevant to the AC.

---

## 1. Global Revisions & New Analyses

### (G1) Quantitative Interpretability Validation (R1-Q1, R2-W1)
We added a human evaluation study (Sec. 4.5; App. F) with **5 PhD-level CS participants** (each with $ \geq 3 $ years AI/ML experience). Cohen’s $ \kappa $ for atom selection:

- **SQA3D:** $ \mu_\kappa = 0.76 $, $ \sigma_\kappa = 0.12 $
- **ScanQA:** $ \mu_\kappa = 0.51 $, $ \sigma_\kappa = 0.13 $

These results show meaningful alignment between ATOM’s decomposition and human reasoning.

### (G2) Multi-Seed Statistical Significance (R1-Q2/Q3, R4-W2)
We added **4-seed experiments** and paired **t-tests** (Sec. 4.1–4.2; Tab. 3):

- **ScanQA:** significant improvement ($ p < 0.01 $)
- **SQA3D:** statistically similar to DSPNet (49.54 ± 0.11% vs. 49.80 ± 0.08%, $ p = 0.098 $)

ATOM therefore maintains competitive SOTA accuracy while providing interpretability.

### (G3) Clarification of the Four Atoms (R2-Q1, R4-Q1)
We clarified that ATOM uses **four** PID-grounded atoms
$ \{ U_1, U_2, R, S \} $
extracted by **three** modules. Respectively, representing 3D uniqueness, 2D uniqueness, redundancy, and synergy.
Notation and descriptions were made fully consistent.

### **(G4) ATOM as an Interpretable Fusion Paradigm (R1-Q4, R4-W2)**
We emphasised that ATOM represents a **paradigm shift** from black-box multimodal fusion to **information-theoretic, interpretable fusion**:

$$
\text{Prior work: } \text{Visual Encoder} \rightarrow \text{Black-box Fusion} \rightarrow \text{Answer}
$$

$$
\text{ATOM: } \text{Visual Encoder} \rightarrow \text{PID-based Fusion} \rightarrow \{\beta, \mathcal{A}\} \rightarrow \text{Answer}
$$

Here $ \mathcal{A} = \{ U_1, U_2, R, S \} $ and $ \beta $ are **interpretable**.
This enables diagnostic analysis, controllable reasoning, and question-type-dependent insight. Please refer to discussion with R1 for finer detail.

### (G5) Atom Weights & Question-Type Patterns (R2-Q2, R3-Q1)
We added extensive analyses (App. E):

- global weight distributions (31,744 samples),
- dataset-specific variation,
- question-type-specific decomposition.

We explained why SQA3D shows cleaner patterns (situated grounding) than ScanQA.

### (G6) Clearer Ablations (R1-W6, R3-Q2, R4-W1)
Tab. 4 was reorganised and moved to the main text.
Reviewer-suggested **$ U_1 $-only and $ U_2 $-only** ablations are in progress and will appear in the final version.

---

## 2. Reviewer-Specific Summary

### R1 (DBXe)
Requested interpretability metrics, significance tests, a clearer description of ATOM’s novelty, and clearer ablations.
All are addressed through the human study (G1), multi-seed tests (G2), clarified paradigm (G4), and reorganised ablations (G6).

### R2 (ifgS)
Asked about atom definitions, “no-atom” baselines, weight ranges, and SID3D.
These are fully clarified in (G3)–(G5), and SID3D is discussed in Related Work.
Many concerns stem from misunderstandings; we encourage a careful re-reading of the revised manuscript.

### R3 (woBS)
Requested improved figure placement, reconciliation of DAM patterns, and $ U_1/U_2 $ ablations—all addressed in (G5)–(G6).

### R4 (pRt9)
Requested clarification of atom definitions, model complexity rationale, and statistical validation.
All addressed via (G2)–(G4), with improved figure readability.

---

## 3. History of Rebuttal & Request to the AC

Important context:

- **R3** explicitly reaffirmed an **accept** recommendation (score 8).
- **R1 and R4** have not yet seen our updated statistical and human-study results.
- **R2** has not engaged since the start of the rebuttal, and many of their concerns reflect misreadings of the original paper. We strongly believe that a careful manual reading is necessary, as all of their points are now thoroughly clarified in the revised manuscript. We respectfully request that R2 review the updated version in full.

---

We kindly ask the AC to examine the updated manuscript and our consolidated responses.

We deeply appreciate the AC’s time and effort.

---

### Meta-Review · Area_Chair_EWNQ · 2026-01-07

**Summary:**

This paper proposes a interpretable framework for 3D question answering. To do this, the paper decomposes multimodal interactions into 3 major steps consisting of view aggregator, a decomposition into four information atoms and a dynamic atom modulation method that upweights atoms to achieve the final answer. Overall, the reviewers were concerned about the complexity of the approach, poor emperical justification of the approach (with many components giving only marginal gains), the poor performance of the method itself on SQA3D, and the lack of the analysis of the individual intepretable elements. After taking a close look at the paper, I agree with these outstanding concerns and do not think the paper should be in the conference in its current form.

**Reviewer Concerns:**

Overall, reviewers were concerned about the extreme complexity of the approach, which was not well justified from emperical results.  In addition, reviewers were concerned about baseline comparisons which were also not well addressed. More minor concerns about text clarifications were addressed in the rebuttal.

**Reviewer Scores:**

I believe that the concerns of both negative reviewers were not adequately addressed in the rebuttal. First, Reviewer ifgS concern about the very large complexity of the approach as well as the absence of recent baselines, is  not addressed. Second, Reviewer DBXe concerns about the weak empirical results, empirical justification of the theor,y as well as the complexity of the approach are also not addressed.

---

### Decision · Program_Chairs · 2026-01-26

Reject